# Ice shelf basal melt rates from a high-resolution DEM record for Pine Island Glacier, Antarctica

David E. Shean[1], Ian R. Joughin[2], Pierre Dutrieux[3], Benjamin E. Smith[2], Etienne Berthier[4]

[1]Department of Civil and Environmental Engineering, University of Washington, Seattle, 98185, USA
[2]Applied Physics Laboratory, University of Washington, Seattle, 98105, USA
[3]Lamont-Doherty Earth Observatory, Columbia University, Palisades, 10964, USA
[4]LEGOS, CNES, CNRS, IRD, UPS, Université de Toulouse, Toulouse, France

*Correspondence to*: David E. Shean (dshean@uw.edu)

**Abstract.** Ocean-induced basal melting is responsible for much of the Amundsen Sea Embayment ice loss in recent decades, but the total magnitude and spatiotemporal evolution of this melt is poorly constrained. To address this problem, we generated a record of high-resolution Digital Elevation Models (DEMs) for Pine Island Glacier (PIG) using commercial sub-meter satellite stereo imagery and integrated additional 2002–2015 DEM/altimetry data. We implemented a Lagrangian elevation change (*Dh/Dt*) framework to estimate ice shelf basal melt rates at 32–256-m resolution. We describe this methodology and consider basal melt rates and elevation change over the PIG ice shelf and lower catchment from 2008 to 2015. We document the evolution of Eulerian elevation change (*dh/dt*) and upstream propagation of thinning signals following the end of rapid grounding line retreat around 2010. Mean full-shelf basal melt rates for the 2008–2015 period were ~82–93 Gt/yr, with ~200–250 m/yr basal melt rates within large channels near the grounding line, ~10–30 m/yr over the main shelf, and ~0–10 m/yr over the North and South shelves, with the notable exception of a small area with rates of ~50–100 m/yr near the grounding line of a fast-flowing tributary on the South shelf. The observed basal melt rates show excellent agreement with, and provide context for, *in situ* basal melt rate observations. We also document the relative melt rates for km-scale basal channels and keels at different locations on the ice shelf and consider implications for ocean circulation and heat content. These methods and results offer new indirect observations of ice-ocean interaction and constraints on the processes driving sub-shelf melting beneath vulnerable ice shelves in West Antarctica.

## 1    Introduction

The Amundsen Sea Embayment (ASE) of the West Antarctic ice sheet (WAIS, Figure 1) has experienced significant acceleration, thinning, and grounding line retreat since at least the 1970s (Joughin et al., 2003; Konrad et al., 2018; Mouginot et al., 2014; Rignot et al., 2014; Rignot, 1998). During this period, regional mass loss increased to present-day estimates of ~100–120 Gt/yr (Medley et al., 2014; Sutterley et al., 2014; Velicogna et al., 2014). These changes appear to be linked to changes in the meridional transport of dense, relatively warm (~0.5–1.2°C, up to +2–4°C above *in situ* freezing point (Jacobs

et al., 2012, 2011; Rignot and Jacobs, 2002)) Southern Ocean sourced Circumpolar Deep Water (CDW) onto the continental shelf (Dutrieux et al., 2014b; Jacobs et al., 1996; Pritchard et al., 2012; Shepherd et al., 2004), where it is funnelled along deep troughs toward the vulnerable grounding lines of large ice streams with reverse bed slopes (Jenkins et al., 2010). Marine ice sheet grounding lines on reverse bed slopes are inherently unstable (Schoof, 2007; Weertman, 1974), and this focused melting

can trigger further grounding-line retreat, acceleration, and dynamic thinning (Joughin and Alley, 2011). Approximately 75% of the West Antarctic Ice Sheet is grounded below sea level, raising concerns about large-scale collapse due to this instability, which could lead to ~3.3 m of global sea level rise (Bamber et al., 2009).

Over the past ~30 years, numerous observational studies have estimated Antarctic ice shelf basal melt rates (e.g., Table S2 of Rignot et al. (2013)). The scope of these efforts ranges from continent-wide remote-sensing inventories (Depoorter et al., 2013;

Paolo et al., 2015; Pritchard et al., 2012; Rignot et al., 2013; Shepherd et al., 2010) to detailed analysis of individual shelves (Berger et al., 2017; Dutrieux et al., 2013; Joughin and Padman, 2003; Moholdt et al., 2014; Wilson et al., 2017). Various methods were used for these assessments, including mass budget ("input-output" or "flux gate") methods (Depoorter et al., 2013; Rignot et al., 2013), satellite laser altimetry (Pritchard et al., 2012), satellite radar altimetry (Paolo et al., 2015; Shepherd et al., 2004), field observations with phase-sensitive radar (Dutrieux et al., 2014a; Jenkins et al., 2006; Langley et al., 2014;

Marsh et al., 2015; Stanton et al., 2013), *in situ* oceanographic observations from autonomous submersibles (Dutrieux et al., 2014b; Jenkins et al., 2010), borehole-deployed instrumentation (Kobs et al., 2014; Stanton et al., 2013), traditional mooring or ship-based oceanographic observations beyond the ice shelf margins (Jacobs et al., 1996, 2011; Jenkins et al., 1997, 2018), and ocean circulation modeling (e.g., Dutrieux et al., 2014b; Payne et al., 2007; Schodlok et al., 2012).

Each of these methods has advantages and disadvantages, differing spatial coverage/resolution, temporal coverage/resolution,

measurement uncertainty, and logistical cost. Many methods require multiple input datasets, and the available data often span different time periods. For example, most previous mass budget analyses combine elevation change rates derived from ICESat altimetry between 2003 and 2008 – a time period characterized by significant change and imbalance in the Amundsen Sea Embayment region – with velocities from a fixed year or a composite mosaic from multiple years (e.g., mosaic of Rignot et al. (2011)). Elevation data from satellite laser and radar altimetry are further limited by large footprints and sparse repeat-track

spacing, with increased uncertainty over areas with non-negligible slopes and/or roughness.

Here, we describe the methods to process and analyze a new dataset of high-resolution DEMs from stereo satellite imagery for Pine Island Glacier (PIG), Antarctica. We use these products to characterize the spatial distribution of ice shelf basal melt and elevation change over the past decade, and evaluate relative melt rates for km-scale ice shelf thickness variations. These methods and results provide a foundation for forthcoming detailed analyses of spatiotemporal evolution of PIG ice shelf basal

melt rates and comparisons with ocean observations.

## 1.1 Pine Island Glacier

Pine Island Glacier (Figure 2) has received significant attention due to the ~30 km grounding line retreat along its centerline (Rignot et al., 2014) (~8 km average retreat across the full width of fast-flowing trunk (Joughin et al., 2016)), ~75% increase

in surface velocity (Mouginot et al., 2014) and >100 m of thinning (Bindschadler, 2002; Pritchard et al., 2009) since the 1970s, with accelerated retreat beginning in the 1990s, likely due to increased ocean heat content, circulation, and basal melt (Jacobs et al., 2011).

Total discharge across the main PIG grounding line increased from ~73 Gt/yr in the mid-1990s to ~114 Gt/yr in 2009 (Mouginot et al., 2014), with a corresponding increase from ~10 to ~12 Gt/yr across the grounding line of the South PIG ice shelf (e.g., the "Wedge" catchment of Medley et al. (2014)). Retreat, speedup and thinning peaked between 2009 and 2010, followed by an observed ~2–3% velocity decrease over the main PIG ice shelf between 2012 and 2013 (Christianson et al., 2016; Mouginot et al., 2014), and return to ~2009 velocities by early 2015. Recent inventories suggest that PIG accounts for nearly ~20% (~120–130 Gt/yr) of present-day West Antarctic discharge and ~40% (40 to 50 Gt/yr) of recent ASE mass loss (Medley et al., 2014; Mouginot et al., 2014; Rignot, 2008). This ice loss corresponds to a sea-level rise contribution of ~0.10–0.15 mm/yr – a substantial portion of the present-day Antarctic Ice Sheet contribution of ~0.2–0.4 mm/yr (Bamber et al., 2018; Church et al., 2013; Rietbroek et al., 2016; The IMBIE team, 2018; WCRP Global Sea Level Budget Group, 2018).

A detailed understanding of the processes (e.g., ocean forcing, marine ice sheet instability) responsible for these observed changes, and their relative importance over time, is critical for future projections of PIG dynamics, mass loss, and contributions to global sea-level rise.

### 1.1.1 Geographic setting

The fast-flowing portion of the PIG ice shelf ("main shelf", Figure 3) is ~25 km wide and nearly 100 km long, with ice thickness of ~1–1.5 km near the main grounding line, and ~300–400 m near the calving front. Surface velocities over the main shelf are currently ~4 km/yr (~11 m/day), with ~2–4-km-wide shear margins that separate the main shelf from the northeast ("North shelf") and southwest ("South shelf') sectors of the PIG ice shelf (Figure 3). In general, surface velocity is relatively slow (<100–500 m/yr) over the North and South shelves, except for a fast-flowing tributary of the South ice shelf with velocity of ~1 km/yr and thickness of ~1 km near the grounding line (Figure 2). Total ice shelf area in recent decades varied from ~5500 to ~6000 km$^2$, due to changes in the grounding line and calving front positions. The PIG catchment (Figure 2A) covers ~1.8–2.0x10$^5$ km$^2$ with annual surface mass balance (SMB) estimates of ~68+/-6 Gt/yr (Medley et al., 2014). The surface of the PIG ice shelf is characterized by a series of longitudinal (approximately along-flow) ridges/troughs near the centerline and transverse (cross-flow) ridges/troughs toward the lateral margins that correspond to basal keels/channels (Vaughan et al., 2012) (Figure 3).

The sub-shelf bathymetry shows a large transverse seabed ridge (TSR) with relief of ~400 m above the adjacent seafloor (Figure 2B and S1). This ridge has been the site of intermittent grounding since the mid-1940s (Smith et al., 2016), and it affects circulation within the cavity, effectively blocking some of the deep, warm CDW from entering the inner cavity (De Rydt et al., 2014; Dutrieux et al., 2014b). We further subdivide the main ice shelf into "inner" and "outer" regions relative to the transverse seabed ridge (Figure S1).

The "ice plain" (e.g., Thomas et al., 2004) mentioned throughout the text describes a region over the inner ice shelf with relatively smooth, gently sloping bed (Figure S1). The lightly grounded "ice plain" was the site of significant grounding line retreat from ~1990s to ~2008, with average rates of ~1 km/yr (Park et al., 2013; Rignot et al., 2014). Our DEM record begins near the end of this retreat, when the "ice plain" region was afloat except for a few isolated grounded spots (Joughin et al., 2016).

### 1.1.2 Oceanographic setting

Westerly surface winds near the continental shelf edge drive northward Ekman transport of surface water away from the continent. This draws deep, relatively warm CDW onto the continental shelf where it flows toward Pine Island Bay along two broad bathymetric troughs carved by previous glacial advances (e.g., Jakobsson et al., 2012; Kirshner et al., 2012). The circulation pathway beneath the PIG ice shelf is less certain, but should generally be clockwise in nature, with modified CDW inflow at depth along the north side of the outer cavity, and outflow of relatively fresh meltwater along the south side of the outer cavity (Dutrieux et al., 2014b). Deep, inflowing water that encounters the large transverse seabed ridge is likely diverted to the south, flowing alongside the ridge within the outer cavity and moving toward the South cavity. Water at intermediate depth is expected to overtop the seabed ridge, creating a sharp density front and a northward jet at the ridge crest (De Rydt et al., 2014; Dutrieux et al., 2014b). Eventually, these waters continue down local bathymetric slopes within the inner cavity toward the grounding line. Once in the inner cavity, the dense, modified CDW reaches the grounding line (Jenkins et al., 2010), with expected cyclonic (clockwise) circulation along the main ice shelf grounding line, and fresh, buoyant meltwater outflow along the centerline and south side of the ice shelf closing the circulation loop. The temporal evolution of this general circulation pattern, and exchange between the inner, outer, and South ice shelf ocean cavities depends on a number of factors, including cavity geometry defined by the evolving ice shelf base and grounding line position.

### 1.1.3 Previous basal melt rate assessments

Recent studies partition the ~2003–2008 PIG mass loss into ~65% (~95–101 Gt/yr) basal melting and ~35% (~50–62 Gt/yr) calving (Depoorter et al., 2013; Rignot et al., 2013), emphasizing the importance of basal melt for this system. Table S2 of Rignot et al. (2013) provides a comprehensive review of past basal melt rate assessments for PIG. Past studies offer a general picture of PIG basal melt rate spatial distribution, with relatively high rates (>100 m/yr) near the main ice shelf grounding line and lower rates over the outer ice shelf (Bindschadler et al., 2011; Dutrieux et al., 2013; Payne et al., 2007). Little is known, however, about basal melt rate temporal variability. Bindschadler et al. (2011) concluded that transverse channels/keels formed annually near the grounding line due to seasonal variability in available ocean heat content (Thoma et al., 2008; Webber et al., 2017), while simulations by Sergienko (2013) showed that similar features may be a spontaneous byproduct of the coupled ice-shelf-plume system with constant ocean heat content.

## 2 Data and methods

We present high-resolution surface elevation observations to investigate the spatial and temporal evolution of PIG. The following sections describe data sources and relevant processing methodology.

### 2.1 Elevation data

We use surface elevation data from a number of sources, including DEMs from satellite stereo imagery, satellite altimetry and airborne altimetry.

#### 2.1.1 WorldView/GeoEye stereo DEMs

We generated DEMs from very-high-resolution commercial stereo satellite imagery (DigitalGlobe WorldView-1, WorldView-2, WorldView-3, and GeoEye-1) using the NASA Ames Stereo Pipeline (ASP, (Beyer et al., 2019, 2018; Shean et al., 2016))

and methodology described by Shean et al. (2016). A total of ~3000 along-track stereopairs from October 2010 to May 2015 were processed for the Amundsen and Bellinghausen Sea coastline of West Antarctica (Figure 1). For this study, we focus our analysis on a ~260x240 km region with dense WorldView/GeoEye DEM coverage covering the PIG ice shelf and lower trunk (Figure 1C).

Stereo image dimensions are typically ~13–17 km wide and 111 km long, with ~0.3–0.5 m ground sample distance (GSD).

The Level-1B (L1B) images were orthorectified using a smoothed version of the Bedmap2 surface DEM (Fretwell et al., 2013) before stereo correlation. For reference, advanced processing settings for ASP included "seed-mode 3" (sparse_disp utility) to initialize the correlation, a 2-level correlation pyramid limit, a correlation timeout of 360 seconds, parabolic sub-pixel refinement, and filtering of isolated disparity map clusters with area <1024 pixels (see Shean et al., 2016 for additional details). We generated additional "cross-track" or "coincident mono" DEMs from pairs of independent mono images with geometry

suitable for stereo reconstruction. We identified candidate pairs in the DigitalGlobe image archive based on the criteria in Table 1, and generated 24 DEMs from images acquired between October 2011 and January 2012. Some of these cross-track pairs were acquired on the same orbit, while others were acquired on different orbits, sometimes by different spacecraft. Final time offsets between the images ranged from 0.007 to 1.6 days.

The cross-track DEMs potentially have increased error due to horizontal displacement errors (i.e., errors due to ice flow

between image acquisitions), non-ideal stereo geometry (e.g., smaller convergence angles) and the fact that some errors in ephemeris data for the two images are independent (as opposed to highly correlated errors for along-track pairs). In practice, these issues can result in increased DEM vertical/horizontal bias and increased relative error (e.g., more "tilt"). Despite potentially increased error, we include these cross-track DEMs in our analysis to fill critical gaps in coverage near the PIG grounding line, and to increase overall DEM sample size for the 2011/2012 season. As described in Section 2.2, these errors

are mitigated through subsequent DEM co-registration and correction.

### 2.1.2 SPIRIT DEMs

We incorporated all six available SPIRIT (SPOT 5 stereoscopic survey of Polar Ice: Reference Images and Topographies, (Korona et al., 2009)) 40-m posting DEMs that covered some portion of the PIG ice shelf between January 5, 2008 and January 18, 2010. Unlike the sub-meter WorldView/GeoEye imagery, the ~5 m GSD SPOT-5 images are unable to resolve meter-scale ice sheet texture, and stereo image correlation often fails for relatively flat, featureless surfaces, leading to gaps in the output DEM. The km-scale ridges/troughs, ~100–1000 m wind-sculpted surface features, and rifts on the main PIG ice shelf, however, provide adequate texture for successful correlation. Compared to the WorldView DEMs, the SPIRIT DEMs include increased noise and additional artifacts, but cover a much larger area (~120 km swath width).

Elevation values in the SPIRIT DEMs are represented as integers, with horizontal and vertical accuracy estimates of <10 m (Bouillon et al., 2006; Korona et al., 2009), which we improve substantially using control points as described in section 2.2.1. We used the DEM V1 products (generated with correlation parameters tuned for gentle slopes), applied the corresponding "CC" mask to preserve correlation scores of 50–100% (masking most interpolated areas), reprojected to a standard Antarctic polar stereographic projection (EPSG:3031), and removed the EGM96 geoid offset to obtain elevations relative to the WGS84 ellipsoid. We filtered the resulting products to remove isolated pixels, mask elevations <20 m above sea level, and remove any pixels with >30 m absolute elevation difference from the per-pixel median of all 2010–2015 WorldView/GeoEye DEMs, effectively removing spurious DEM values associated with clouds in the original imagery.

### 2.1.3 Satellite and airborne altimetry

The NASA Operation IceBridge (OIB) mission collected airborne altimetry data over PIG during annual campaigns from 2009/2010 to 2014/2015, except for the 2013/2014 season. Most campaigns occurred during October–November, with data acquisition flights for a particular site typically occurring over ~1–3 days. We assembled all available NASA Airborne Topographic Mapper (ATM, (Krabill et al., 2002; Martin et al., 2012)) and Land, Vegetation, and Ice Sensor (LVIS, (Blair et al., 1999; Hofton et al., 2008)) airborne lidar data for use in our study area. A total of 25 ATM flights and 7 LVIS flights crossed the study area during the period from 2009 to 2015, with data collection for each flight typically lasting <4 hours. The high-altitude LVIS surveys on October 20, 2009 and October 10, 2011 covered a significant portion of the main ice shelf, while other LVIS/ATM flights generally consisted of a few sparse flightlines distributed across the ice shelf.

We processed all altimetry data as described by Shean et al. (2016), and produced gridded 32-m and 256-m DEMs with sparse coverage for each campaign using the ASP point2dem utility. This utility assigns the output value for each grid cell by computing the weighted mean of all points within a 1-grid-cell-width radius.

We also included available 2003–2009 NASA ICESat Geoscience Laser Altimeter System (GLAS, (Schutz et al., 2005; Zwally et al., 2002)) satellite altimetry data. These data were clustered by ~33-day campaign and gridded as described above, providing 18 additional sparse DEMs. While caution must be exercised during interpretation of these sparse data over rough surfaces or steep slopes, we included them in our analysis to extend the observational record back to 2003.

## 2.2 DEM co-registration and correction

The following sections describe a cascading co-registration and correction workflow used to improve both absolute and relative
DEM accuracy over the PIG study area.

### 2.2.1 Co-registration with altimetry

Where possible, a point-to-point iterative closest point (ICP) algorithm (Shean et al., 2019, 2016) was used to co-register
DEMs to filtered altimetry data from the sources described in Section 2.1.3. The altimetry data were queried for each DEM
extent and the returned points were limited to "static" (e.g., nunataks) and "dynamic" (e.g., slow-moving ice with limited
slope/roughness) control surfaces. We removed points with time offset between the altimetry point timestamp and DEM
timestamp ($|t_{altimetry} - t_{DEM}|$) of >1 year. Any points over floating portions of the PIG ice shelf were excluded. The
remaining points were further filtered using a maximum expected displacement (product of measured surface velocity
magnitude and time offset between the point and DEM timestamp) threshold of 10 m. All control points were assumed to have
vertical accuracy of ~0.1 m (see section 5.1 of Shean et al., 2016).
The majority of the WorldView/GeoEye DEMs had $10^6$–$10^8$ filtered points available for co-registration, with
$|t_{altimetry} - t_{DEM}|$ of only a few months. The ICP co-registration provided translation corrections for 368 of 575 DEMs over
the PIG catchment, with a significant improvement in multiple quality metrics following co-registration (Figure 4, Table 2).
Uncorrected DEMs had an initial mean vertical bias of +3.1 m above the altimetry data (Figure 4), as discussed in section 6.1.1
of Shean et al. (2016), and we applied a -3.1 m vertical correction to the remaining 207 DEMs that lacked adequate control
data.

The filtered SPIRIT DEMs were co-registered with the ICP routine described in Section 2.2.1, and the results are shown in
Figure S2. In addition to the filtered airborne data, a large sample of near-contemporaneous ICESat GLAS data were available
for co-registration of the 2008–2009 SPIRIT DEMs. After co-registration we estimate that the lower-resolution SPIRIT DEM
products have 3–4 m or better absolute vertical accuracy (1-sigma). One of the DEMs (January 3, 2009) had large residual
offsets between control point and DEM elevation, and we performed a secondary round of vertical bias correction (-3.1 m) to
minimize offsets between this DEM and a 2010–2015 WorldView/GeoEye DEM per-pixel median elevation composite over
flat, smooth surfaces near the main ice shelf.

### 2.2.2 Elevation correction for ocean and atmospheric variability

After DEM co-registration, we corrected all elevation data (including altimetry) over the floating portions of the PIG ice shelf
to remove the effects of ocean tides, atmospheric pressure (Inverse Barometer Effect (IBE), (e.g., Padman et al., 2003)) and
mean dynamic topography.

We computed tidal amplitude $\Delta h_t$ using the CATS2008A inverse barotropic tide model (an updated version of the model
described by Padman et al. (2002)). The inverse barometer effect magnitude $\Delta h_{IBE}$ was computed from 6-hour interval ERA-

Interim mean sea level pressure reanalysis data (Dee et al., 2011). We removed the 2002–2016 median pressure (985.21 hPa), and scaled residuals by ~1 cm/hPa to obtain the approximate inverse barometer correction. Tidal amplitude for DEM timestamps ranged from -0.75 to +1.04 m ($\sigma = 0.33$ m), while the inverse barometer effect amplitude ranged from -0.3 to 0.35 m ($\sigma = 0.11$ m) (Figure S3). These high-frequency (hourly–daily) corrections show good agreement with observed surface elevation records from GPS receivers on the PIG ice shelf (Shean et al., 2017).

The mean dynamic topography ($\Delta h_{MDT}$) correction removes residual offsets between the geoid and mean sea level due to ocean circulation. Estimates for mean dynamic topography near ASE are approximately -1.2 m (Andersen and Knudsen, 2009). Corrected ice surface elevation above sea level is calculated as:

$$h = h_e - \Delta h_g - \alpha(\Delta h_{MDT} + \Delta h_t + \Delta h_{IBE}) \tag{1}$$

Where $h_e$ is measured elevation above the WGS84 ellipsoid and $\Delta h_g$ is the EGM2008 geoid offset (Pavlis et al., 2012) (approximately -27.6 to -24.4 m across PIG ice shelf). To provide a smooth transition from grounded to freely floating ice, we defined the coefficient $\alpha$ to increase linearly with distance $l$ downstream of the grounding line:

$$\alpha(l) = \begin{cases} 0, l \le 0, \\ 0.33l, 0 < l \le 3 \; km, \\ 1, l > 3 \; km. \end{cases} \tag{2}$$

For this study, the grounding line (Figure 2 and 3) was defined with a single composite polygon derived from DInSAR (Joughin et al., 2016; Rignot et al., 2014) and high-resolution DEM data, with an approximate timestamp of 2011.

After correction using Equation (1), surface elevation from airborne altimetry approaches 0 m above sea level over open water. We neglect elevation change due to long-term sea level rise (~0.3 cm/yr) and glacial isostatic adjustment (elastic response is approximately +2–4 cm/yr for ASE (Barletta et al., 2018; Groh et al., 2012; Gunter et al., 2014)).

## 2.2.3    WorldView/GeoEye DEM "tilt" correction

As identified by Shean et al. (2016), a subset (~5–10%) of the WorldView/GeoEye DEMs appear to have a slight along-track and/or cross-track "tilt" of ~1–3 m over the ~111 km strip length, likely due to small errors in spacecraft attitude metadata. For most of these "tilted" DEMs, the available control point spatial distribution is insufficient to constrain a rigid-body ICP rotation. Initial attempts using bootstrapping and least-squares minimization of offsets between adjacent, overlapping DEMs to solve for a "tilt correction" failed due to overfitting and the propagation of larger errors near some DEM edges. To correct these problematic DEMs, we developed an optimization approach that simultaneously solved for interannual *dh/dt* and planar corrections to remove individual DEM tilt. In principle, this is similar to the SERAC method used for altimetry over the Greenland ice sheet (Csatho et al., 2014; Schenk and Csatho, 2012).

The WorldView DEM record (November 16, 2010 to April 6, 2015) postdates the period of rapid PIG speedup that ended in ~2009, and surface velocities and SMB display limited variability from 2010–2015 (Christianson et al., 2016; Shean et al., 2017). Thus, while the dynamic response to earlier rapid grounding line retreat and speedup continues to propagate upstream across the PIG catchment, we expect relatively limited variability in elevation change rates during this period.

We manually masked the main ice shelf and fast-flowing grounded ice stream within ~30 km of the grounding line, and then used the criteria listed in Table 3 to identify "dynamic control surfaces" (Figure 5) over grounded ice with limited linear trend (*dh/dt*) and limited residual variance about this trend. Over these surfaces, the elevation at any particular DEM pixel $i$ (with spatial coordinates $x_i$ and $y_i$) at time $j$ is given by:

$$h_{i,j} = (a_i t_j + b_i) + (c_j x_i + d_j y_i + e_j) \qquad \textbf{(3)}$$

where $a_i$ and $b_i$ represent the slope and offset of a linear model fit to elevation values at pixel $i$, and coefficients $c_j$, $d_j$ and $e_j$ define a planar correction for all $i$ within a DEM at time $t_j$.

We solved for these coefficients using least-squares minimization with regularization and a smoothness constraint designed to penalize large spatial gradients. Elevation values from filtered, gridded altimetry data were included in the solution with increased weight. Stereo DEMs with <40 km along-track length were limited to a vertical offset correction ($e_j$), with no tilt correction ($c_j = d_j = 0$). Limits for tilt magnitude were increased for cross-track DEMs (Section 2.1.1) and limits for vertical offset were increased for input DEMs that were not initially co-registered using ICP. Tilt magnitude was limited in the DEM cross-track direction, as most of the observed tilt was in the DEM along-track direction. Figure 5 and Table 4 summarize the results of these corrections, with considerable improvement in all metrics.

### 2.2.4    Output elevation data

We prepared a resampled "stack" of all co-registered, corrected DEMs over the PIG ice shelf using a common 256-m grid. Additional stacks with increased grid resolution (64-m and 32-m, respectively) were prepared over high-priority areas such as the inner ice shelf and GPS validation sites (Shean et al., 2017)).

### 2.3    Post-correction DEM accuracy

As discussed by Shean et al. (2016), the uncorrected vertical/horizontal accuracy for the along-track stereo DEMs is <5.0 m. After systematic artifact removal and co-registration, vertical accuracy can be less than <0.2–0.4 m for surfaces with <10° slope. For the PIG ice shelf, we conservatively estimate the final DEM accuracy to be ~1 m after co-registration and least-squares "tilt" correction. We initially expect increased uncertainty for 2013/2014 DEMs due to reduced availability of OIB altimetry data during this season. This uncertainty, however, was reduced after the least-squares correction, which leveraged altimetry data and corrected WorldView/GeoEye DEMs from adjacent years.

Several factors can reduce the effectiveness of DEM co-registration with altimetry. The primary problems for PIG include sparse control data with limited variation in surface slope and aspect, and longer $\left| t_{altimetry} - t_{DEM} \right|$ time offsets (~1–12 months). Over these timescales, surface processes (e.g., accumulation/ablation, wind redistribution of snow) can potentially lead to surface elevation changes of ~1 m, and advection of small-scale surface features can lead to horizontal co-registration errors.

We used a network of five 2012–2014 GPS sites on the outer ice shelf (Shean et al., 2017) as independent check points for WorldView DEMs. Corrected DEM elevations show good agreement (~0.72 m root mean squared error [RMSE] and ~0.57 normalized median absolute deviation [NMAD]) with cm-accuracy surface elevations derived from GPS interferometric reflectometry (GPS-IR) antenna height records at each site. Unfortunately, no valid SPIRIT DEM pixels were available near the 2008–2010 GPS sites.

## 2.4 Annual surface elevation composites and mosaics

We generated weighted-average composites using the ASP dem_mosaic utility for all available elevation data in a given year (September–April, but typically October–March), with a nominal January 1 timestamp (Figure 6). For each output pixel, the weighted averaging algorithm assigns greater weight to input pixels from spatially continuous sources (e.g. DEMs with few data gaps) and penalizes isolated pixels or clusters of pixels (see ASP documentation for details). The resulting composites appear seamless, but can include smoothing artifacts due to variable temporal sampling of input elevation data, especially for features that advect in the along-flow direction.

Adjacent WorldView/GeoEye stereo images are often acquired weeks or months apart during a particular season due to clouds and/or competition for resources. Even after DEM co-registration and correction, this asynchronous sampling can introduce horizontal and vertical feature offsets between adjacent DEMs in fast-flowing regions. Generally, this sampling is not a problem for smaller targets covered by a single WorldView/GeoEye DEM footprint (e.g., Greenland outlet glacier termini). Larger targets like the PIG ice shelf, however, require >10 WorldView/GeoEye DEMs for complete coverage, and more sophisticated mosaicking approaches are necessary to preserve local features.

To obtain full ice shelf coverage while also preserving timestamps and relative elevation values within individual input DEMs, mosaics without averaging or blending were generated for the ~October-March period each year. We used a "reverse" ordering scheme for input DEM timestamps, so that the last DEM from each season was mosaicked on top. Finally, we generated WorldView/GeoEye DEM mosaics when complete ice shelf coverage was available over a relatively short time span (e.g., October–December 2012, Figure 3). In such cases, input DEM products were manually selected and ordered to minimize feature offsets.

## 2.5 Surface velocity

Surface velocity data constrain horizontal ice shelf advection rates and aid interpretation of observed elevation change. In an effort to generate self-consistent velocity and DEM products, we estimated velocity using feature tracking with normalized cross-correlation of two DEMs, similar to the approach described by Dutrieux et al. (2013). However, this approach is susceptible to spurious correlations and data gaps over flat, featureless areas, especially for low-resolution inputs (e.g., 40-m SPIRIT DEMs). This technique also fails for longer time intervals (>2 years), as surface processes, deformation, rotation due to velocity gradients, and spatially variable basal melt decreased coherence. For these reasons, we used an independent set of

gridded velocity products, which enabled reconstruction of particle paths for arbitrary elevation data, including sparse altimetry.

We compiled 22 surface velocity mosaics (Christianson et al., 2016; Joughin, 2002; Joughin et al., 2010) from TerraSAR-X/TanDEM-X, Advanced Land Observing Satellite (ALOS) Phased Array type L-band Synthetic Aperture Radar (PALSAR) and Landsat-8 (LS8) data (Figure 2C). The 500-m ALOS and LS8 products cover the entire PIG ice shelf during late 2006, 2007, 2008, 2010, 2013, 2014, and 2015, while the 100-m TSX/TDM products are available every ~3–6 months over the main ice shelf from 2009–2015.

We derived spatially and temporally continuous velocity fields for the full PIG ice shelf using piecewise linear interpolation via 3-D $(x,y,t)$ Delaunay Triangulation. Linear barycentric interpolation was then used to extract spatially continuous velocity grids with 512-m resolution for a regular time interval of 122 days from January 1, 2008 to June 1, 2016. The interpolated velocity products were smoothed in the time dimension with a 610-day, 2$^{nd}$-order Savitzy-Golay filter, and then in the spatial dimension with a 2.5-km rolling median filter to mitigate artifacts in the input mosaics. To obtain velocity fields with increased

spatiotemporal sampling, we performed secondary interpolation with a high-resolution timestep (e.g., 5–20 day) and increased spatial sampling (e.g., 32–256 m), with a final Gaussian smoothing filter (~0.17-km sigma) applied in the spatial dimension to reduce any residual interpolation artifacts. The basal melt rate calculations described in Section 3.2 required estimates of the velocity divergence, which we calculated from these interpolated, smoothed velocity products for each high-resolution time step using a central-difference approach.

**2.6    Bed topography**

We evaluated five different bed datasets for PIG (Figure S1), including Bedmap2 (Fretwell et al., 2013), an aerogravity inversion constrained by Autosub bathymetric data (De Rydt et al., 2014; Dutrieux et al., 2014b), an aerogravity/Autosub inversion constrained by active-source seismic surveys (Muto et al., 2016), a mass-conserving bed embedded in Bedmap2 (Morlighem et al., 2011), and the CReSIS L3 gridded Multichannel Coherent Radar Depth Sounder (MCoRDS) ice thickness

product from 2009–2010 airborne radio echo sounding. The extent and resolution of these products is variable, with significant elevation differences (>100–300 m) in places, especially over the PIG inner cavity (Figure S1).

We produced a new combined bed dataset (Figure S1C) using aerogravity/Autosub data, existing open-water bathymetry, and all available quality-controlled CReSIS MCoRDS and British Antarctic Survey (BAS) Polarimetric Airborne Survey Instrument (PASIN) ice thickness measurements collected over grounded ice. We used "anisotropic interpolation" to fit a

smooth surface to these data using an inversion procedure that preferentially minimizes bed curvature in the along-flow direction, while matching the bed elevation at data points to within the estimated data errors (see methods of Medley et al., 2014; Mueller et al., 2012). While some local "peaks" over the longitudinal seabed ridge beneath the PIG ice shelf may be biased high, this bed appears most consistent with observed recent grounding line evolution (Joughin et al., 2016).

## 2.7 Surface mass balance (SMB)

The Regional Atmospheric Climate Model (RACMO) v2.3 (Ettema et al., 2009; Lenaerts et al., 2012; Van Meijgaard et al., 2008; Van Wessem et al., 2014) provides continent-wide estimates of surface mass balance on a 27-km grid. To estimate SMB over the PIG ice shelf, we used monthly average SMB products available through December 2013, and repeated the observed 2013–2014 SMB signal for calculations spanning 2014–2015. We generated gridded RACMO SMB products with the same extent and spatial sampling as the DEM and velocity products using bicubic interpolation.

## 3 Elevation change and basal melt rate derivation

We consider elevation change for PIG using both Eulerian *dh/dt* (fixed reference grid) and Lagrangian *Dh/Dt* (grid moving with the surface) descriptions. These two approaches are complementary and provide distinct information over grounded and floating ice.

### 3.1 Theory

Assuming incompressibility, constant ice density, and column-average velocity $\mathbf{u}$, the Eulerian description of mass conservation for a column of ice with ice-equivalent thickness $H$ can be expressed as:

$$\frac{\partial H}{\partial t} = -\nabla \cdot (H\mathbf{u}) + \dot{a} - \dot{b} \tag{4}$$

where $\dot{a}$ is surface mass balance (meters ice equivalent for time interval $dt$) and $\dot{b}$ is basal melt rate (meters ice equivalent, defined as positive for melt).

The flux divergence term, $\nabla \cdot (H\mathbf{u})$, can be expanded as:

$$\nabla \cdot (H\mathbf{u}) = H(\nabla \cdot \mathbf{u}) + \mathbf{u} \cdot (\nabla H) \tag{5}$$

where $\nabla \cdot \mathbf{u}$ is the velocity divergence (positive for extension) and $\nabla H$ is the thickness gradient.

The relationship between Lagrangian (denoted by material derivative operator $\frac{D}{Dt}$) and Eulerian thickness change is provided by the material derivative definition:

$$\frac{DH}{Dt} = \frac{\partial H}{\partial t} + \mathbf{u} \cdot (\nabla H) \tag{6}$$

Equations 4, 5, and 6 can be combined to obtain Lagrangian thickness change for the column:

$$\frac{DH}{Dt} = -H(\nabla \cdot \mathbf{u}) + \dot{a} - \dot{b} \tag{7}$$

Over grounded ice, we assume that the bed elevation remains constant, and can substitute Eulerian surface elevation change
*dh/dt* for Eulerian thickness change *dH/dt*. This assumption does not hold for floating ice. If we assume hydrostatic equilibrium, however, we can estimate freeboard ice thickness from observed surface elevation. We remove firn-air content $d$ from observed

surface elevation $h$ to obtain ice-equivalent freeboard surface elevation, and then compute ice-equivalent freeboard thickness $H_f$:

$$H_f \approx (h - d)\left(\frac{\rho_w}{\rho_w - \rho_i}\right) \tag{8}$$

assuming a constant density for sea water ($\rho_w$) and ice ($\rho_i$). This ice-equivalent freeboard thickness $H_f$ can then be substituted for $H$ in Equation 7. We assume that any changes in $d$, $\rho_w$ and $\rho_i$ are negligible during our study period, so the *DH_f/Dt* term reduces to Lagrangian surface elevation change (*Dh/Dt*), resulting in a modified mass-conservation expression for a column of floating ice:

$$\frac{Dh}{Dt} = -(h - d)(\nabla \cdot \mathbf{u}) + \left(\dot{a} - \dot{b}\right)\left(\frac{\rho_w - \rho_i}{\rho_w}\right) \tag{9}$$

### 3.2 Eulerian long-term interannual trend

To characterize long-term (~5–10 year) elevation change over the PIG ice shelf during and after the period of rapid grounding-line retreat, we computed interannual per-pixel trends for the 2003–2010 and 2010–2015 periods. These trends were determined using a linear fit to surface elevation for each grid cell with 3 or more observations, with >6 valid samples available for most cells. No smoothness constraint was imposed – all fits were computed independently, although adjacent elevation values are highly correlated.

### 3.3 Basal melt rate

Both Eulerian and Lagrangian frameworks can be used to estimate the basal melt rate. The Lagrangian description tracks elevation change for the same column of ice over time, eliminating potential aliasing due to advection of high-frequency surface gradients (i.e., ice shelf surface ridges and troughs). If velocity divergence and surface mass balance are known, Equation 9 can be rearranged to solve for the component of observed elevation change due to basal melt:

$$\dot{b} = -\left(\frac{Dh}{Dt} + (h - d)(\nabla \cdot \mathbf{u})\right)\left(\frac{\rho_w}{\rho_w - \rho_i}\right) + \dot{a} \tag{10}$$

### 3.4 Basal melt rate implementation

Past studies of basal melt rate using a Lagrangian framework used *in situ* observations (e.g., Jenkins et al., 2006), a single pair of gridded DEM observations (e.g., Dutrieux et al., 2013), or a series of sparse altimetry data (e.g., Moholdt et al., 2014). The approach presented here uses hundreds of independent DEM observations with variable spatial coverage over an 8-year time period. This set of DEMs provides thousands of combinations for basal melt rate computation, with the flexibility to vary the time interval *Dt*. Most of the PIG ice shelf DEM data were acquired seasonally from ~October–March, so we computed interannual *Dh/Dt* for time intervals of ~1 and ~2 years. Longer time intervals decrease spatial resolution, as the observed

*Dh/Dt* values are integrated across a longer path, but they provide improved signal-to-noise ratio for *Dh/Dt*, and we use the ~2 year products for further analysis.

To calculate basal melt rate, we track each pixel in an earlier DEM acquired at time $t_i$ (*DEM$_i$*) to its corresponding downstream location where it intersects a later DEM acquired at time $t_j$ (*DEM$_j$*). Since our velocity fields vary over time (Section 2.5), an

appropriate time step $\Delta t$ for this tracking is automatically determined based on the grid cell size and maximum velocities (e.g., ~10–20 days for 256-m grid). For each time step $n$ $\{n \mid 0 < n \leq Dt/\Delta t, n \in \mathbb{Z}^+\}$, all valid pixels ("particles") from *DEM$_i$* are propagated along flow paths (Figure 7) computed from the time-variable velocity fields. This propagation yields updated *DEM$_i$* particle positions at time $(t_i + n\Delta t)$. For those particles whose paths intersect *DEM$_j$*, we calculate the observed Lagrangian elevation change rate as:

$$\frac{Dh}{Dt} = \frac{h_j - h_i}{t_j - t_i} \qquad \qquad \textbf{(11)}$$

The observed cumulative particle *Dh/Dt* is then used to estimate evolving surface elevation $h$ at each time step $n$ along the particle path (assuming the *Dh/Dt* rate is constant), and local velocity divergence $(\nabla \cdot \mathbf{u})$ values are sampled at each time step $n$ along the particle path from the continuous velocity products described in Section 2.5. The corresponding local $h(\nabla \cdot \mathbf{u})$ is then integrated over the full path:

$$h(\nabla \cdot \mathbf{u}) \approx \frac{\sum_{n=0}^{Dt/\Delta t} \left(h_i + n\Delta t \frac{Dh}{Dt}\right)(\nabla \cdot \mathbf{u})_n}{Dt} \qquad \qquad \textbf{(12)}$$

This approach should accurately capture time-variable thinning/thickening due to local velocity divergence experienced along

each path, rather than sampling velocity divergence from single, fixed velocity grid. We also sampled time-variable SMB grids at each time step, but the spatiotemporal variability for the monthly 27-km products is limited along the ~8 km particle paths, and we used a time-averaged estimate for $\dot{a}$ extracted at the particle path midpoint. Finally, we substituted the cumulative particle *Dh/Dt* and local $h(\nabla \cdot \mathbf{u})$ into Equation 10, which provides an integrated basal melt rate estimate for a single pixel across a single pair of DEMs.

### 3.5 Basal melt rate path distribution

We consider two end members for the spatiotemporal distribution of ice shelf basal melt rates. End member #1 assumes a fixed, 3D "melt rate field" in the ocean cavity beneath the PIG ice shelf that varies spatially but not temporally, so that features with variable draft (i.e., keels and channels) melt at different rates as they advect through this field. End member #2 assumes that melt rate spatial variability is highly correlated with local ice shelf thickness gradients (and associated basal slope), so that

local melt rates advect with features on the ice shelf (e.g., once formed, a transverse basal channel will continue to melt at a similar rate as it advects downstream). In reality, basal melt rates are likely sensitive to some combination of these two end-member scenarios.

The methodology described in Section 3.4 provides basal melt rate estimates for each particle in a Lagrangian reference frame. For subsequent analysis on a regular grid, we must remap these observations into a common, global Eulerian reference frame.

This step is complicated by the fact that the long time intervals between DEM observations (~2 years) and high advection rates (~4 km/yr) on the main PIG ice shelf result in particle path lengths (~8 km) that greatly exceed the input DEM grid cell size and the desired melt rate grid cell size (256 m). To address this issue and to evaluate the two basal melt rate end-member scenarios, we developed two approaches to work with basal melt rates from Lagrangian *Dh/Dt* measurements in an Eulerian reference frame: "along-flow distribution" and "initial-pixel" (Figure 7).

### 3.5.1 Along-flow distribution

The "along-flow distribution" approach partitions observed particle basal melt rates (Section 3.4) evenly across each path, and computes statistics for each cell in a fixed Eulerian grid using all paths that pass through that cell (Figure 7). This approach potentially provides a more realistic map of the melt rate "field" (end member #1), but it effectively smooths basal melt rate estimates in the along-flow direction, especially for longer path lengths. This leads to reduced resolving power for local basal 425 melt rate spatial variability (end member #2), especially for features with transverse orientation (e.g. transverse channels/keels, rifts).

The path history of all valid particles for a particular $DEM_i$–$DEM_j$ combination is reduced to identify a unique set of occupied grid cells in the global Eulerian reference frame. For each particle path, basal melt rate $\dot{b}$ is calculated as described in Section 3.4 and these values are distributed evenly along encountered cells. This procedure yields a spatially variable particle count 430 within each cell in the global Eulerian coordinate system; only one particle will pass through a cell on the upstream edge of the domain, while ~10–100 particles could pass through a cell near the center of the domain over the full *Dt* interval. We then compute the median and NMAD for each cell (Figure 7). This approach reduces noise and provides metrics to evaluate variance and uncertainty in derived basal melt rates.

### 3.5.2 Initial-pixel

The "initial-pixel" approach assigns particle basal melt rate values to the corresponding path origins in $DEM_i$, so the resulting basal melt rates grids have the same spatial extent as $DEM_i$. This approach is relatively straightforward, and was used in earlier work (e.g., Dutrieux et al., 2013). It preserves the relative spatial distribution of basal melt rates across individual features in $DEM_i$ (e.g., channels and keels), but doesn't resolve where along the ~8 km particle path that melt actually occurred.

For a given $DEM_i$–$DEM_j$ combination, the initial-pixel approach assigns particle basal melt rate values to $DEM_i$ pixel locations. 440 For each initial $DEM_i$, we then create stacks of available $DEM_i$–$DEM_j$ initial-pixel basal melt rate products and compute a per-pixel stack median map. In other words, basal melt rates calculated from each valid downstream $DEM_j$ are assigned to the initial $DEM_i$ pixel locations, and median values for each $DEM_i$ pixel are computed assuming no temporal variability in basal melt rates for all valid $t_i$-$t_j$ intervals.

### 3.5.3 Path distribution considerations

Under melt-rate end-member #1, the initial-pixel approach will introduce a negative bias for a fixed basal melt rate field with relatively large negative spatial gradient (e.g., 200 m/yr to 100 m/yr over 8 km in the inner cavity), as the mean path basal melt rate (150 m/yr) will be assigned to the initial-pixel locations (where rates are locally 200 m/yr). We experimented with an approach using path mid-point locations rather than initial-pixel locations, but this resulted in large gaps near the grounding line and prevented direct comparison of basal melt rates with the original $DEM_i$ elevations. Under melt-rate end-member #2, the initial-pixel approach provides more realistic basal melt rate magnitude and spatial distribution than the along-flow distribution approach. The difference between the two approaches will be negligible for areas of the PIG ice shelf with low surface velocity (<250 m/yr).

## 3.6 Basal melt rate composites

In the above sections, we described basal melt rate calculations for a single $DEM_i$–$DEM_j$ combination with sufficient overlap and a $t_i$-$t_j$ time interval that falls within the chosen $Dt$ range (~2 years), which represents only one of many potential valid $DEM_i$–$DEM_j$ combinations that can be formed from the full set of DEMs over the PIG ice shelf.

For a given $DEM_i$, after we calculate basal melt rates using the first viable $DEM_j$, the $DEM_i$ particles are further propagated and the process is repeated for all other viable $DEM_j$ until the $t_i$-$t_j$ time interval exceeds the maximum $Dt$ interval. The entire process is then repeated for all possible $DEM_i$.

For our chosen $Dt$ of ~2 years, a total of 117 unique $DEM_i$ with initial $t_i$ timestamps spanning 2008–2013 and sufficient $DEM_j$ intersection area were available over the PIG ice shelf. Each $DEM_i$ formed ~2–40 valid $DEM_i$–$DEM_j$ combinations, yielding a final set of >1000 independently generated $DEM_i$-$DEM_j$ basal melt rate products.

The individual $DEM_i$-$DEM_j$ basal melt rate products can have relatively high uncertainty and/or limited spatial extent, so we created annual melt-rate mosaics and composites to reduce noise and increase total spatial coverage. We used different methodology for the "along-flow distribution" and "initial-pixel" approaches, as described below.

### 3.6.1 Along-flow distribution composites

We generated weighted-average basal melt rate composites from individual "along-flow distribution" basal melt rate products. This approach provides basal melt rate grids centered on January 1 for the ~2-year interval products. For each grid cell in the output mosaics, the weighted-average approach favors pixels near the center of input products with larger areal coverage. Per-pixel standard deviation is also calculated for each ~2-year basal melt rate composite, providing maps that capture the spatial distribution of basal melt rate uncertainty (and any true basal melt rate temporal variability during the ~2-year period). The annual composites were then used to generate a mean basal melt rate composite for the full 2008–2015 period.

### 3.6.2 Initial-pixel mosaics

The per-pixel stack median products described in Section 3.5.2 provide high-resolution maps of local basal melt rates, but they are limited to the *DEM$_i$* spatial extent. To overcome this limitation, we generated mosaics of the stack median products using a reverse time-ordering scheme, so basal melt rate estimates for the most recent *DEM$_i$* timestamp were mosaicked on top. This approach preserves the local basal melt rate distribution within each stack median product, while providing coverage over as much of the ice shelf as possible, with limited time offset between spatially adjacent observations. These products can be directly compared with surface elevation (and corresponding freeboard thickness estimates) from the reverse time-order DEM mosaics described in Section 2.4.

### 3.7 Uncertainty and sources of error

Surface elevation uncertainty over the PIG ice shelf includes errors due to the geoid model (~0.1–0.4 m), mean dynamic topography (~0.2 m), and tide/IBE correction (~0.1 m). For simplicity, we assume a constant firn-air content of 12 m with uncertainty of +/-2 m to account for any spatial and temporal variability (see Appendix of Shean, 2016). We used a depth-averaged density for ice and underlying ocean water of 917 +/-5 kg/m$^3$ and 1026 +/-1 kg/m$^3$, respectively, and assume that these densities are constant in both space and time. We assume uncorrelated errors of 1 m for surface elevation, 50 m for bed elevation, 30 m/yr for velocity (for ~37.5° look angle and +/-0.5 m tide) (Joughin, 2002) and 28% for SMB (Depoorter et al., 2013).

Our conversion from surface elevation to ice thickness assumes that the ice shelf is in hydrostatic equilibrium (Shean et al., 2017). We use a consistent methodology and the same assumptions of hydrostatic equilibrium for the full 2008–2015 study period, which increases confidence in observed temporal change. We do not update the grounding line mask for basal melt rate calculations, and some of the persistent high and low basal melt rate values <1–2 km downstream of the grounding line may be related to evolving grounding line position and insufficient masking over grounded ice (Joughin et al., 2016; Milillo et al., 2017). Transient regrounding of keels will yield increased surface elevations and larger apparent freeboard thickness values. This may also lead to localized ice deformation and non-zero vertical strain rates that are inconsistent with the assumption that surface velocity equals the column-average velocity.

Uncertainty for elevation change and basal melt rate products depends on the time interval. For example, assuming that errors are uncorrelated, a 1-m absolute error in surface elevation should result in ~1.4 m combined error in elevation change. This elevation change uncertainty should remain constant, so integrating observations over longer periods will result in greater signal-to-noise for annual elevation change rates (e.g., ~1.4 m/yr error for a 1-year interval or ~0.7 m/yr for a 2-year interval, assuming constant rates). This estimate does not, however, include slope-dependent vertical error due to cumulative horizontal displacement error, which will increase for longer time intervals. It is challenging to quantify this *Dh/Dt* uncertainty contribution in a forward sense, as multiple sources (e.g., cumulative displacement error from velocities, DEM co-registration, DEM resampling) can lead to slope- and aspect-dependent errors. Basal melt rate products can also include artifacts over shear

margins and near the ice front due to anomalously large *Dh/Dt* values (+/-20–40 m) from advection of near-vertical surface gradients (e.g., ice front, icebergs, rifts) and errors in velocity divergence.

The stacking and averaging approaches described in Section 3.6 should reduce many of these errors, but this improvement is difficult to capture with formal error estimates. The initial-pixel stack per-pixel NMAD (Section 3.5.1) and along-flow per-pixel standard deviation (Section 3.6.1) metrics can provide maps of uncertainty, but these estimates will also include any true
basal melt rate temporal variability during the observation period.

## 4    Results

### 4.1    Long-term Eulerian elevation change

Figure 8 shows long-term Eulerian elevation change (*dh/dt*) for the full study area. From 2003 to 2010, thinning rates <30 km upstream of the grounding line were ~5–10 m/yr, while those farther upstream over the catchment were ~1 m/yr. From 2010
to 2015, thinning rates near the grounding line decreased to ~0–1 m/yr, with increased thinning of ~1–2 m/yr over the catchment. Thinning rates also increased to ~3–4 m/yr over upstream ice stream shear margins within ~60 km of the grounding line, especially the north shear margin.

A series of curvilinear elevation anomaly "bands" with orientation approximately transverse to flow is apparent over the catchment ~40–100 km upstream of the grounding line (Figure 8D). These features are related to dense series of arcuate surface
crevasses (e.g., Scott et al., 2010) that display elevation change due to advection. Individual DEMs show elevation differences of ~0.5 m between these crevasse bands and inter-band surfaces.

Over the PIG ice shelf, we observe 2010–2015 *dh/dt* signals with spatial scales of ~10–15 km that are unrelated to advection of km-scale surface features (Figure 8D). We observe ~1–2 m/yr thickening downstream of the grounding line on the north side of the inner main ice shelf and ~1 m/yr thinning over the south side of the outer main ice shelf. The South PIG ice shelf
shows <1 m/yr thinning from 2010–2015, with ~3 m/yr thinning over upstream ice within ~10 km of the grounding line. The North ice shelf shows little elevation change with <0.5–1 m/yr thinning upstream of the grounding line.

### 4.2    Basal melt rate spatial distribution

Figure 9 shows mean 2-year basal melt rate products for the 2008–2015 period. Full ice shelf basal melt rates were ~82 Gt/yr for "initial-pixel" and ~93 Gt/yr for "along-flow distribution" composite 2-year products.
In general, basal melt rates are >150–200 m/yr near the main ice shelf grounding line, with highest rates of >250 m along the north side of the grounding line (Figure S4). Basal melt rates are generally ~50–100 m/yr over the main ice shelf inner cavity, where ice thickness exceeds ~600–700 m, and ~10–30 m/yr over most of the outer ice shelf, where ice thickness is ~300–500 m. We observe considerable anisotropy, with longitudinal spatial correlation over lengths scales of ~20 km and significant transverse ~km-scale variability. This is true for both the "initial-pixel" and "along-flow distribution" products (Figure 9),
suggesting that this anisotropy is not a result of smoothing in the along-flow direction. The northern third of the outer main ice

shelf displays ~3–4 longitudinal features with elevated basal melt rates of ~30–40 m/yr (red arrow in Figure 9). Upstream of these features, a broad (~10 km wide x 20 km long) region of low-relief transverse ridges/troughs displays reduced basal melt rates of ~5–10 m/yr (green arrow).

Basal melt rates are ~0–10 m/yr over the South ice shelf and ~0–5 m/yr over the North ice shelf (Figure 9). High basal melt rates of ~60–90 m/yr are observed near the relatively deep (~900 m) grounding line of the fast-flowing (~0.7–1.0 km/yr) South ice shelf tributary. Elevated basal melt rates of ~20–50 m/yr are also observed within large channels on the South ice shelf (blue arrow in Figure 9). Integrated basal melt rates over the North and South ice shelves are ~5 and ~10 Gt/yr, respectively.

## 4.3    Channel-scale melt distribution

We used the "initial-pixel" basal melt rate mosaics to evaluate observed basal melt rates for basal channels and keels on the Main ice shelf. We applied a high-pass filter (1.5 km sigma Gaussian) to annual "reverse" order DEM mosaics (Section 2.4), and defined masks for channels and keels using filtered elevations less than -1 m and greater than +1 m, respectively (Figure 10 and S5). These masks were applied to corresponding 2-year "initial-pixel" basal melt rate products, and weighted-average composites were generated from all available years to document the spatial distribution of main ice shelf channel and keel melt rates for the 2008–2015 period. The value at any given pixel in the channel (keel) composite is derived from melt rates for several advecting channels (keels) that intersected that pixel over time, providing a sample of background melt rates (end member #1 in Section 3.5) for channel (keel) features at different locations in the cavity.

The highest basal melt rates are associated with longitudinal surface ridges (basal keels) within ~3–4 km of the grounding line. In the inner cavity (~4–15 km from the grounding line), high basal melt rates (>100 m/yr) are associated with both longitudinal surface troughs (basal channels) and surface ridges (basal keels). Several persistent channels display high basal melt rates throughout the 2008–2015 record, but there is more apparent temporal variability associated with deep keels due to grounding and ungrounding.

Over the mid to outer ice shelf, we observe relatively high basal melt rates on keels (~20–40 m/yr) and limited basal melt rates in transverse channels (~0–10 m/yr). Both channels and keels display higher basal melt rates over the northern portion of the outer ice shelf (red arrow in Figure 9). Higher basal melt rates of ~10–20 m/yr are observed over ~50–70 km-long longitudinal keels near the ice shelf centerline, while ~0 m/yr basal melt rates are observed within adjacent longitudinal channels. One prominent longitudinal keel displays basal melt rates of ~30–40 m/yr (black arrow in Figure 9).

## 5    Discussion

## 5.1    Long-term elevation change

Grounding line retreat and speedup through 2010, combined with inherent marine ice sheet instability, are primarily responsible for the strong thinning observed upstream of the grounding line at PIG (Joughin et al., 2010). Our observations show that this thinning decreased after 2010 (Figure 8), which is consistent with results from model simulations documenting

the inland migration of the associated speedup (Joughin et al., 2010). The end of rapid grounding line retreat and the regrounding of deep keels on the transverse seabed ridge (Christianson et al., 2016; Joughin et al., 2016) likely contributed to decreased thinning rates immediately upstream of the grounding line after 2010. The continued thinning over upstream shear margins (Figure 8) can also be attributed to this evolution, as sustained thinning rates of >5–10 m/yr over the main trunk prior to 2010 (Flament and Rémy, 2012; Joughin et al., 2010; Wingham et al., 2009) led to an increase in surface slopes and transverse driving stress across the shear margins.

## 5.2    Basal melt rate spatial distribution

Our results show a ~11 Gt/yr difference between the full ice shelf along-flow distribution and initial-pixel basal melt rate estimates, with most of this difference over the inner cavity. This discrepancy is likely related to large spatial gradients in the "fixed" melt rate field (end member #1), which we would expect to introduce a negative bias in the initial-pixel basal melt rate estimates, as described in Section 3.5. Thus, the along-flow distribution melt-rate of ~93 Gt/yr estimate is likely a better full ice shelf estimate. The along-flow distribution and initial-pixel basal melt rates are comparable on the outer ice shelf and slow-moving areas of the North and South ice shelves, with both offering good resolution of basal melt rates for longitudinal surface features (e.g., channels and keels).

The spatial distribution of high basal melt rates near the grounding line (Figure S4) is likely a function of modern (post-2006) cavity geometry (Figure S1) and sub-shelf circulation. Mass conserving bed reconstruction for the 1990s configuration revealed a large longitudinal seabed ridge (~4 km wide x 30 km long) near the centerline of the inner cavity (Rignot et al., 2014). The highest basal melt rates of >200–250 m/yr are observed on the north side of this longitudinal seabed ridge, where warm, salty water circulating at depth through the inner cavity first reaches the grounding line (e.g., Dutrieux et al., 2014b). The enhanced ~30–40 m/yr basal melt rates over the northern portion of the outer ice shelf (red arrow in Figure 9) are located immediately downstream of the transverse seabed ridge (Figure S1). Both the Autosub observations and ocean GCM simulations show increased ocean current velocity and enhanced variability due to cold water intrusion near this location (Dutrieux et al., 2014b), suggesting that this local high in basal melt rates could be related to local circulation patterns and/or upwelling. This location is also one of the expected pathways for warm CDW inflow into the inner cavity (e.g., St-Laurent et al., 2015), and we suggest that as this water flows over the transverse seabed ridge, it could lead to enhanced turbulence, vertical heat transport towards the ice base, and increased basal melting.

## 5.3    Channel-scale melt distribution

Our results are generally consistent with past work (e.g., Dutrieux et al., 2013) suggesting that higher melt rates are associated with basal channels in the inner cavity, and basal keels over the outer ice shelf (Figure 10). Inner-cavity channels/keels have much higher relief than outer ice shelf channels/keels, so we might expect higher basal melt rates due to faster plume-driven flow along inner-cavity channels. However, our results also show high basal melt rates over deep keels in the inner cavity,

especially within ~5 km of the grounding line (Figure S5), suggesting that high heat content and local circulation may dominate basal melting at these depths.

Our results demonstrate the potential for high-resolution Lagrangian $Dh/Dt$ measurements of channel-scale features on ice shelves, even with known methodological limitations (see Section 2.10; discussion in Dutrieux et al., 2013; Shean et al., 2017). Keels on the mid to outer PIG ice shelf typically reach water depths up to ~400–450 m, while channels are typically ~300–350 m. These features should intersect the observed thermocline, with temperature gradients of over 1.0°C possible between ~300 and ~450 m depth (Dutrieux et al., 2014b). Our results are consistent with the hypothesis that enhanced melting of outer

ice shelf keels is related to their exposure to warmer water at depth (end member #1 in Section 3.5), with reduced plume-driven flow in the channels due to limited ice thickness gradients. The transverse surface ridges and troughs on the south side of the main ice shelf display greater relief than those along the north side of the ice shelf (Figure 3), with correspondingly higher basal melt rates over the deeper keels (Figure 10). Based on these preliminary results, we suggest that analysis of keel melt rates over time could provide new information about the spatiotemporal evolution of the thermocline in the outer cavity.

**5.4    Comparison with past basal melt-rate assessments**

The local basal melt rates observed near the grounding line within the deep inner cavity (>200 m/yr, Figure 9 and S4) are significantly higher than some past estimates of ~100 m/yr from observations (Bindschadler et al., 2011; Dutrieux et al., 2013) and ~70-120 m/yr from ocean circulation modeling (Dutrieux et al., 2014b; Payne et al., 2007; Seroussi et al., 2014). They are more consistent with flux divergence melt rate estimates of ~200–300 m/yr near the mid-1990s grounding line by Payne et al.

(2007), and ~200 m/yr near the 2009 grounding line by Dutrieux et al. (2013).

Our full ice shelf mean basal melt rates for the period between 2008 and 2015 (~82–93 Gt/yr) are less than, but within the reported uncertainty of past estimates for the period between 2003 and 2008: 95+/-14 (Depoorter et al., 2013) and 101+/-8 Gt/yr (Rignot et al., 2013). While it is possible that no change occurred between the 2003–2008 period and the 2008–2015 period, the apparent decrease in mean melt rate would be consistent with melt rate estimates from oceanographic observations

of ~100 Gt/yr in 2007 to ~73 Gt/yr in 2009 (Dutrieux et al., 2014b). However, this apparent decrease may be at least partially attributable to methodological differences between our study and previous studies (e.g., ice shelf area, flux-gate placement). The previous studies also mixed observations from different time intervals during a highly dynamic period in PIG's recent history, with $dh/dt$ from ICESat data acquired between 2003 and 2008, velocities from an InSAR mosaic with approximate timestamp of 2007–2008 (Rignot et al., 2011), and average SMB for the period from 1979 to 2010. Furthermore, these studies

relied on interpolation of sparse ICESat tracks to estimate spatially continuous Eulerian $dH/dt$ for the entire PIG ice shelf (e.g., -5.32+/-0.3 m/yr (Rignot et al., 2013)). The ICESat GLAS laser spot was ~30–70 m in diameter with ~170 m along-track spacing and ~20 km cross-track spacing between repeat tracks over PIG (e.g., Figure 3 of Pritchard et al. (2009)). Limited measurements were available to constrain local slopes sampled by repeat ICESat tracks over the PIG ice shelf, and aliasing of advecting km-scale surface ridges and troughs can lead to significant errors in thinning rates inferred from smoothed ICESat

repeat tracks (e.g. Figure 13 of Sergienko (2013)), especially after converting inferred elevation change to freeboard thickness

change. While this may not be relevant for relatively flat, smooth ice shelves with high ICESat track density like the Ross and Ronne-Filchner ice shelves (e.g., Moholdt et al., 2014), this issue complicates analysis of the sparse ICESat *dh/dt* measurements over the relatively rough PIG ice shelf, and previous uncertainty estimates for full ice shelf basal melt rates based on ICESat observations are likely too low. Thus, while basal melt rates may have been higher between 2003 and 2008, we cannot rule out the possibility that no long-term change occurred between the 2003–2008 and 2008–2015 periods. Observations with ~20-km spatial resolution (e.g., ICESat or radar altimetry (e.g., Paolo et al., 2015)) can capture the long-term temporal evolution of Eulerian elevation change and basal melt for the full PIG ice shelf, but they cannot directly capture changes associated with dynamic ice-ocean processes that operate on shorter spatial scales. The high-resolution DEM record and methodology presented here allows for both full ice shelf basal melt rate estimates and analysis of the detailed spatiotemporal evolution of km-scale features that are coupled to sub-shelf circulation and local basal melting. As the high-resolution DEM record for Antarctica continues to grow, future analyses for PIG and other Antarctic ice shelves will provide new insight into the underlying processes controlling ice-ocean interaction, with implications for future ice sheet stability.

## 6  Summary/conclusions

We developed a method to correct and integrate high-resolution DEM observations with satellite/airborne altimetry data and surface velocity data to estimate Eulerian *dh/dt*, Lagrangian *Dh/Dt*, and ice shelf basal melt rates. Mean 2008–2015 basal melt rates for the full PIG ice shelf were ~82–93 Gt/yr. Local basal melt rates were ~200–250 m/yr near the grounding line, ~10–30 m/yr over the outer main ice shelf, and ~0–10 m/yr over the North and South ice shelves, with notable exception of ~50–100 m/yr near the grounding line of a fast-flowing tributary on the South ice shelf. The basal melt rates from Lagrangian *Dh/Dt* measurements show excellent agreement with, and provide spatial/temporal context for, *in situ* basal melt rate observations. Basal melt rates vary substantially across ~km-scale ice shelf thickness variations, with greater melting observed in basal channels and over deep keels near the grounding line and relatively shallow keels over the outer ice shelf. The methods and general results presented here provide a foundation for further analysis of the detailed spatiotemporal evolution of basal melt rates and connections with ocean observations for the PIG ice shelf during the 2008–2015 period.

## Code/Data availability

The Level-1B DigitalGlobe images used to generate DEMs were provided by the Polar Geospatial Center at the University of Minnesota, under the NGA NextView License. The NASA Ames Stereo Pipeline code and binaries used to generate the DEMs from these images are available from https://ti.arc.nasa.gov/tech/asr/groups/intelligent-robotics/ngt/stereo/. Derived data products will be made available upon request. Code used for data processing and analysis is available from https://github.com/dshean and/or will be made available upon request.

## Author contribution

DS led conceptualization, data curation, formal analysis, funding acquisition, investigation, methodology, resources, software, validation, visualization, and writing.

IJ supported conceptualization, funding acquisition, methodology, project administration, resources, supervision, and writing.

BS supported conceptualization, methodology, software, supervision, and writing.

PD supported data curation, methodology, validation, and writing.

EB supported data curation (provided SPIRIT DEMs) and writing.

## Competing interests

The authors declare that they have no conflict of interest.

## Acknowledgements

D. Shean was supported by NASA NESSF fellowship award NNX12AN36H. I. Joughin was supported by NASA Cryosphere awards NNX15AD54G and NNX17AG54G and NSF OPP award 1643285. B. Smith was supported by NASA Cryosphere award NNX13AP96G. P. Dutrieux was supported by NSF OPP award 1643285 and the UW Applied Physics Laboratory. E. Berthier was supported by the French Space Agency through the TOSCA program. We acknowledge Claire Porter, Paul Morin, and others at the Polar Geospatial Center (NSF ANT-1043681) who managed tasking, ordering, and distribution of the L1B commercial stereo imagery under the NGA NextView license. We thank Oleg Alexandrov, Zack Moratto and Scott McMichael for additional development of the Ames Stereo Pipeline with support from the NASA Cryosphere program. We thank Stefan Ligtenberg and Peter Kuipers Munneke for providing RACMO SMB products. Howard Conway and Nick Holschuh offered productive comments on an earlier version of this manuscript. Resources supporting this work were provided by the NASA High-End Computing (HEC) Program through the NASA Advanced Supercomputing (NAS) Division at Ames Research Center. The SPOT-5 DEMs and imagery were provided at no cost by the French Space Agency (CNES) through the SPIRIT International Polar Year project (Korona et al., 2009). We thank two anonymous reviewers for their thoughtful comments and suggestions, which improved this manuscript.

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

## Tables

| Convergence angle | 10-60° |
|---|---|
| Time between images | <2 days |
| Minimum intersection area | 100 km$^2$ |
| Minimum relative image area for intersection | 30% |

**Table 1: Cross-track stereo pair criteria.**

| | Along-track Stereo | | Cross-track stereo | | Combined | |
|---|---|---|---|---|---|---|
| Count | 343 | | 25 | | 368 | |
| Mean offset before/after co-registration (m) | -3.06 | -0.01 | -4.03 | 0.02 | -3.12 | -0.01 |
| Mean RMSE before/after (m) | 3.29 | 0.44 | 5.24 | 0.73 | 3.42 | 0.46 |
| Mean NMAD before/after (m) | 0.36 | 0.36 | 0.63 | 0.63 | 0.38 | 0.38 |

**Table 2: Statistics for elevation difference between WorldView/GeoEye DEMs and altimetry control points, before and after DEM co-registration.**

| Minimum number of DEMs | 4 |
|---|---|
| Minimum total dt | 1.5 years |
| Minimum elevation (EGM2008) | 10 m |
| Maximum absolute dh/dt | 2.0 m/yr |
| Maximum detrended std | 3.0 m |

**Table 3: Criteria to identify dynamic control surfaces for least-squares DEM correction. See Figure 5 for resulting map of dynamic control surfaces.**

| | Mean (m) | | Median (m) | |
|---|---|---|---|---|
| | Std. | Detr. Std. | Std. | Detr. Std. |
| Original | 2.45 | 2.11 | 2.49 | 2.08 |
| Co-registered | 1.29 | 0.78 | 0.94 | 0.56 |
| Co-registered and LS "tilt" correction | 1.14 | 0.41 | 0.73 | 0.22 |

**Table 4: Results of least-squares DEM correction. Statistics computed for 2010-2015 WorldView/GeoEye DEMs and ATM/LVIS altimetry data over dynamic control surfaces (n=4–44 at each pixel, sample of ~6.1x10$^5$ pixels, covering ~4x10$^4$ km$^2$). All metrics show decreased spread after correction, with median values less prone to outliers.**

**Figures**

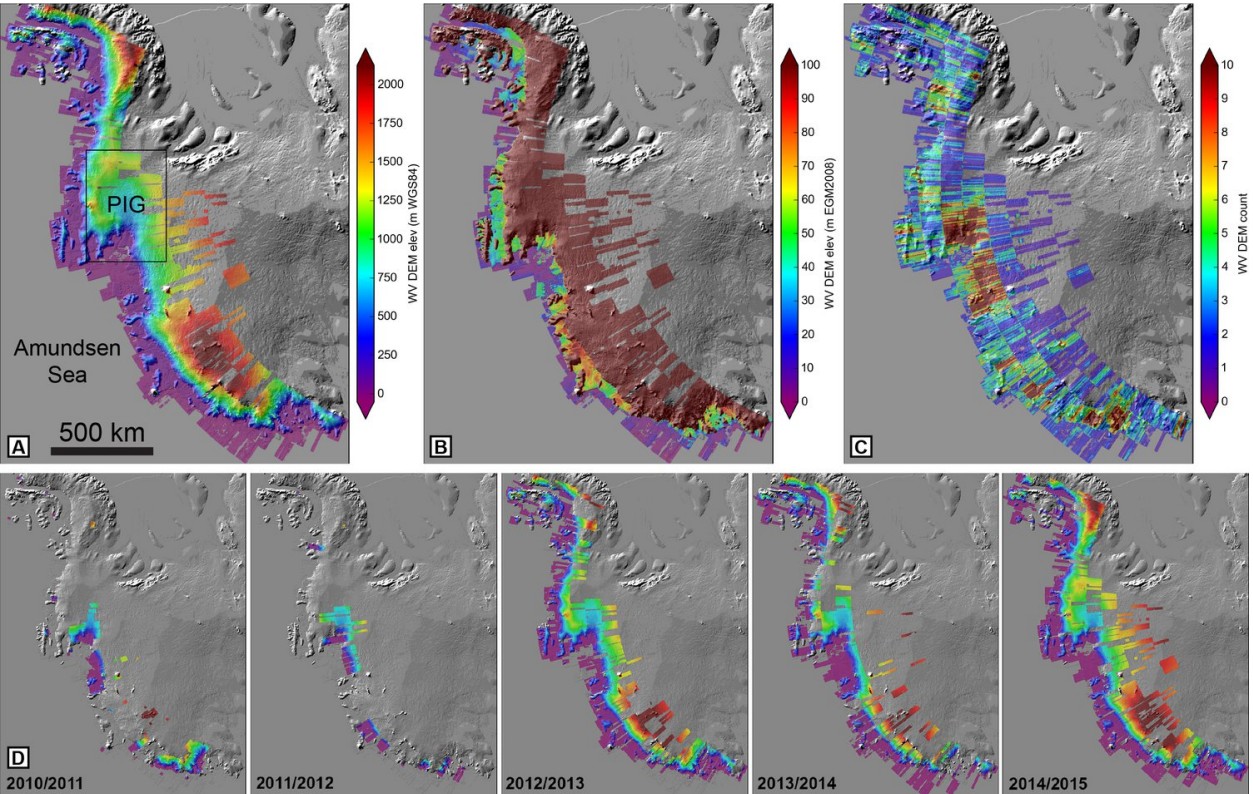

**Figure 1: Cumulative and annual DEM composites for West Antarctica. A) Weighted-average composite of ~3000 WorldView/GeoEye stereo DEMs from 2010-2015, overlaid on Bedmap2 shaded relief map. Total cumulative 2-m DEM coverage is 4.1 million km². Black box shows location of Figure 2. B) DEM composite with elevation values relative to EGM2008 geoid (approximates mean sea level) and color stretch to show surface elevation of floating ice shelves. C) Total count of DEMs for the 2010-2015 time period and D) Annual DEM mosaics with same color scale as in panel A. Note increased annual coverage over time, with good coverage of PIG ice shelf in all years. Projection is Antarctic polar stereographic (EPSG:3031).**

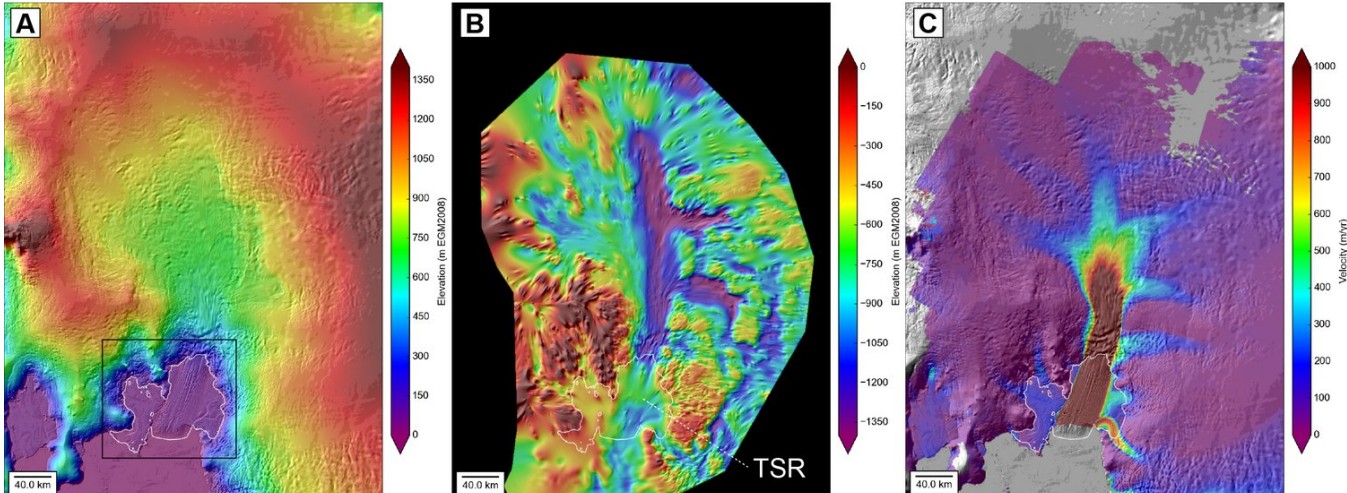

**Figure 2: Context for the PIG catchment: A) High-resolution WorldView/GeoEye DEM mosaic over Bedmap2 DEM. White outline shows PIG ice shelf and ~2011 grounding line. Black box shows location of Figure 3. B) Combined bed topography and bathymetry from anisotropic interpolation of radar-derived ice thickness and other sources (see Section 2.6). Note bedrock channels beneath main trunk and tributaries. Dotted white line shows location of transverse seabed ridge (TSR) in PIG ice shelf cavity (see Figure S1 for detailed bed intercomparison). C) Median 2006-2016 surface velocity magnitude with color ramp saturated at 1 km/yr to show detail over tributaries (see Figure 1 of Shean et al. (2017) for color ramp saturated at 4 km/yr).**

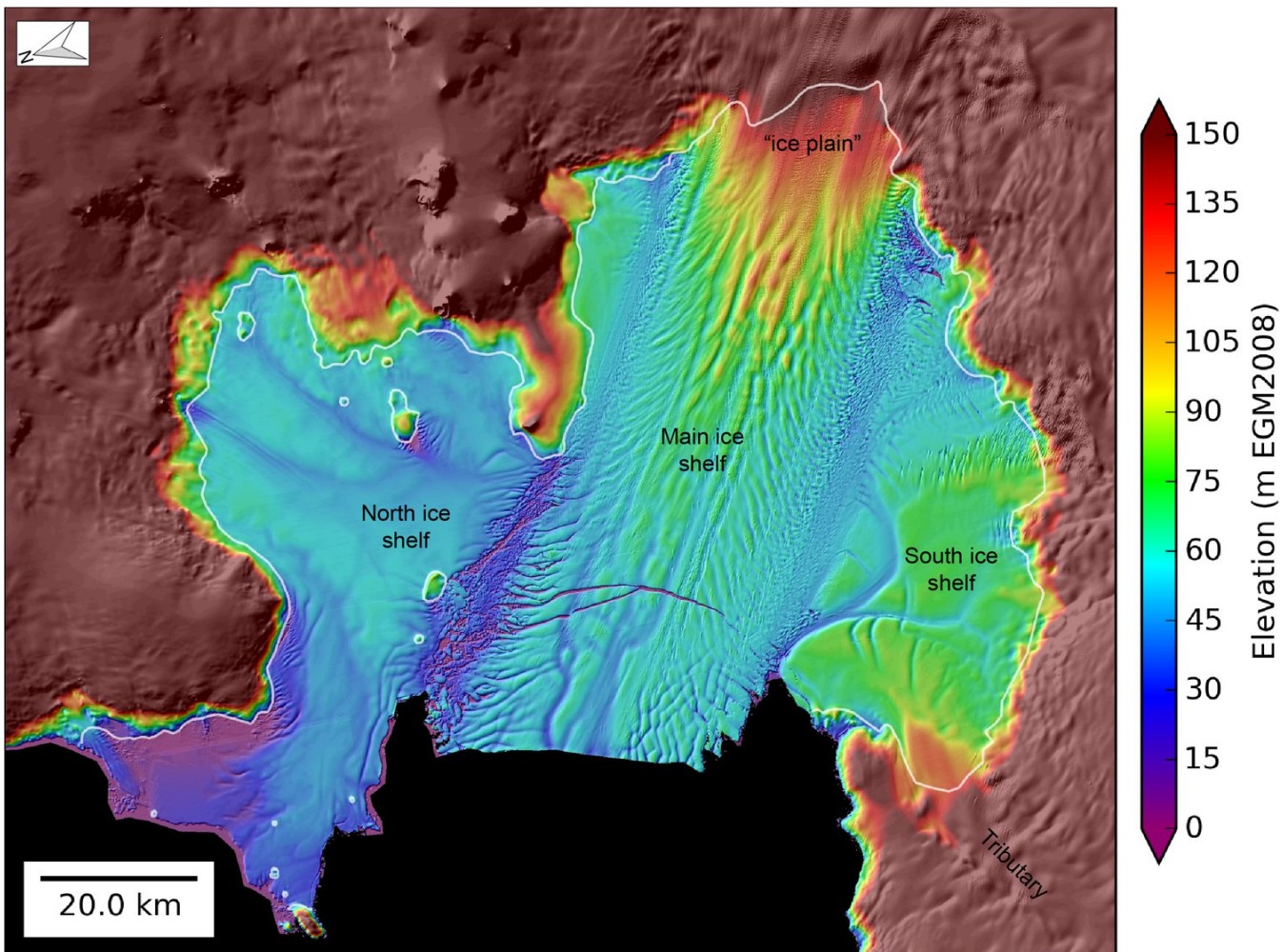

**Figure 3: October–December 2012 WorldView/GeoEye DEM mosaic of the PIG ice shelf. Labels show regions discussed in text: North ice shelf, South ice shelf, Main ice shelf, "ice plain", and fast-flowing South ice shelf tributary. White outline shows ~2011 grounding line. Elevation values are corrected surface height (Equation 1) above the EGM2008 geoid.**

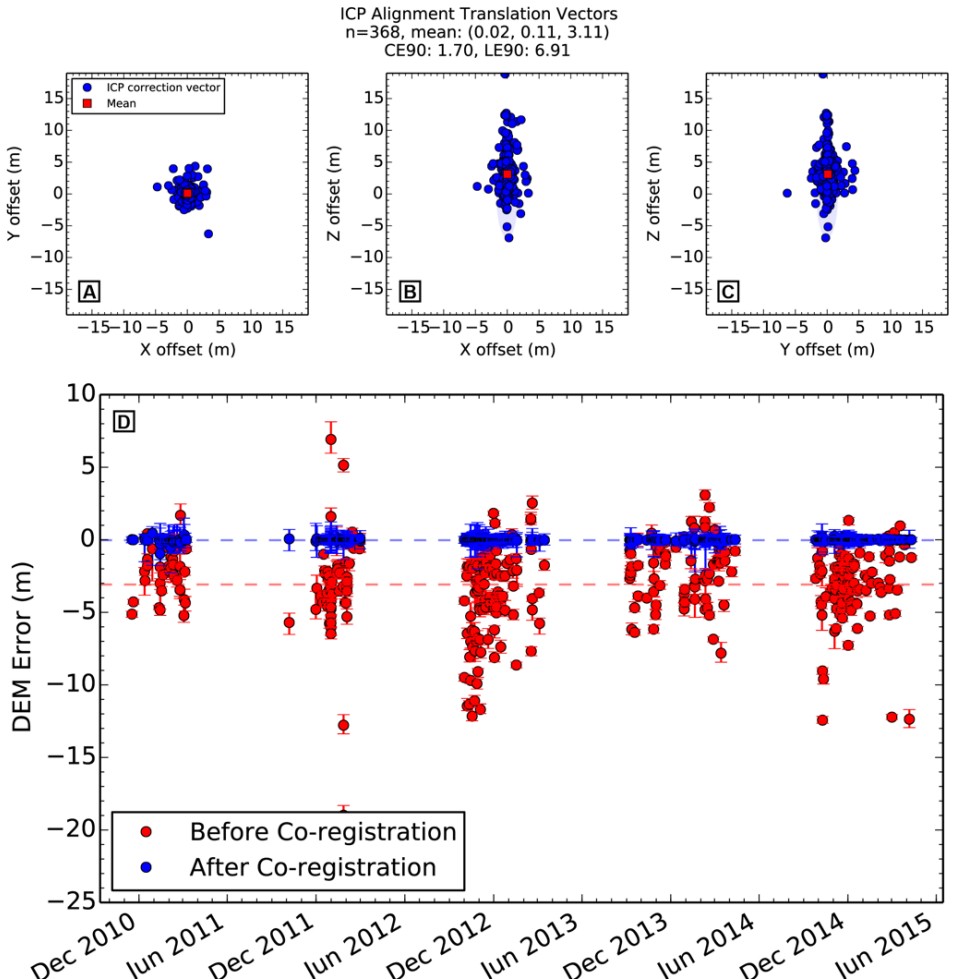

**Figure 4: Co-registration results for 368 WorldView/GeoEye DEMs over PIG catchment (see Shean et al. (2016) for additional details). A-C) Iterative Closest Point (ICP) translation vector components required to co-register each DEM with filtered altimetry data. D) Median DEM error (DEM - altimetry) with error bars showing 16-84% spread for each DEM, before (red) and after (blue) co-registration. Horizontal dashed lines show mean error values. The 2011/2012 cross-track stereo DEMs display larger errors before co-registration. After co-registration, bias is removed and residual error spread for individual DEMs is typically <0.5–1 m, as summarized in Table 2.**

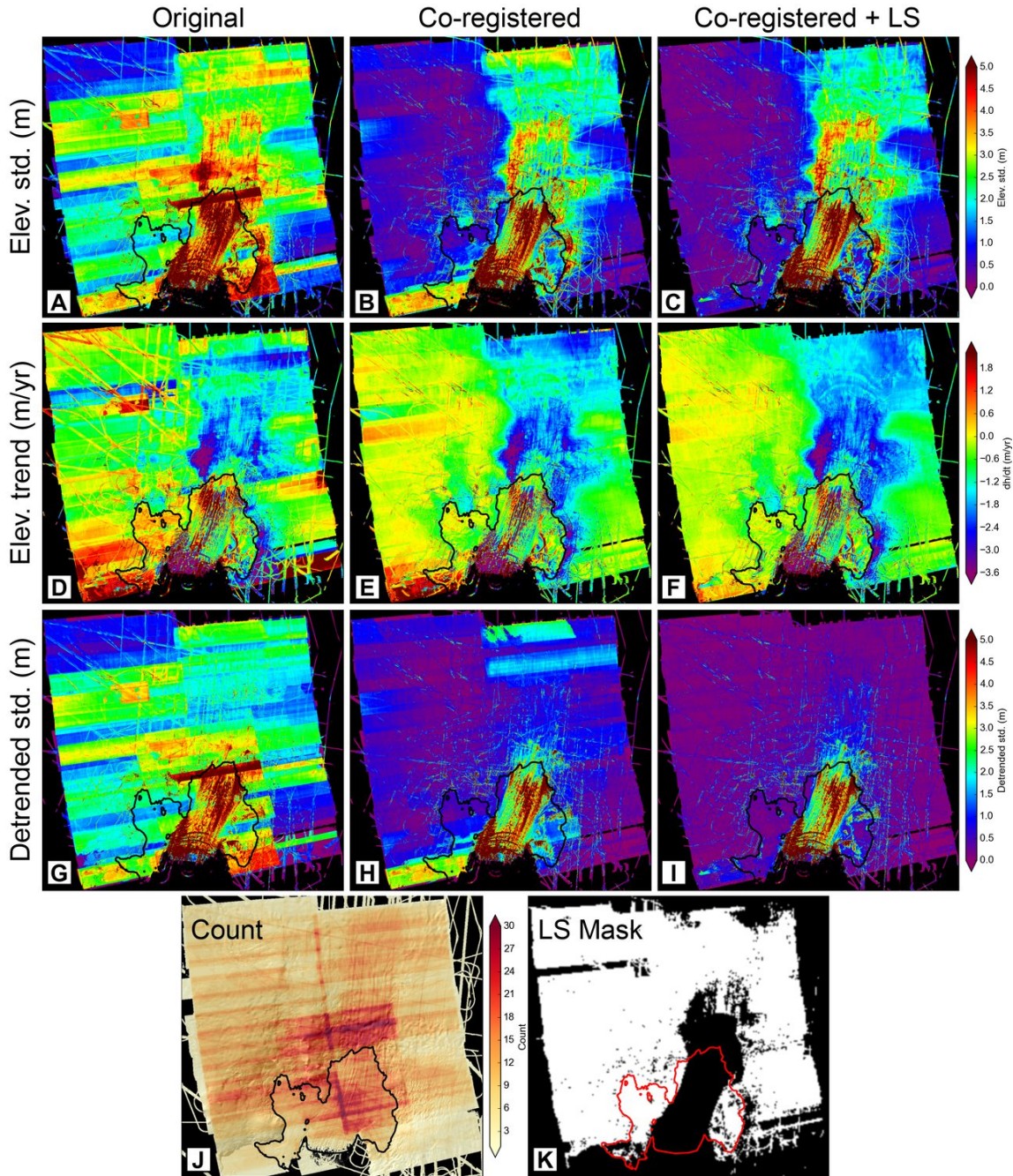

**Figure 5: Statistics for 2010-2015 WorldView/GeoEye DEMs and available 2009-2015 ATM/LVIS altimetry data over the PIG study area. Top row (A-C) shows per-pixel elevation standard deviation, second row (D-F) shows per-pixel linear elevation trend, and third row (G-I) shows per-pixel standard deviation of residuals from linear regression. Left column (A,D,G) shows values for original DEM products before correction, center column (B,E,H) after ICP co-registration to filtered altimetry data, and right column (C,F,I) after least-squares optimization to correct residual DEM "tilt". Note overall improvement of final correction (right column). Bottom row shows per-pixel DEM count (J) and dynamic control surfaces (white) used during least-squares correction (K), as defined by criteria in Table 3.**

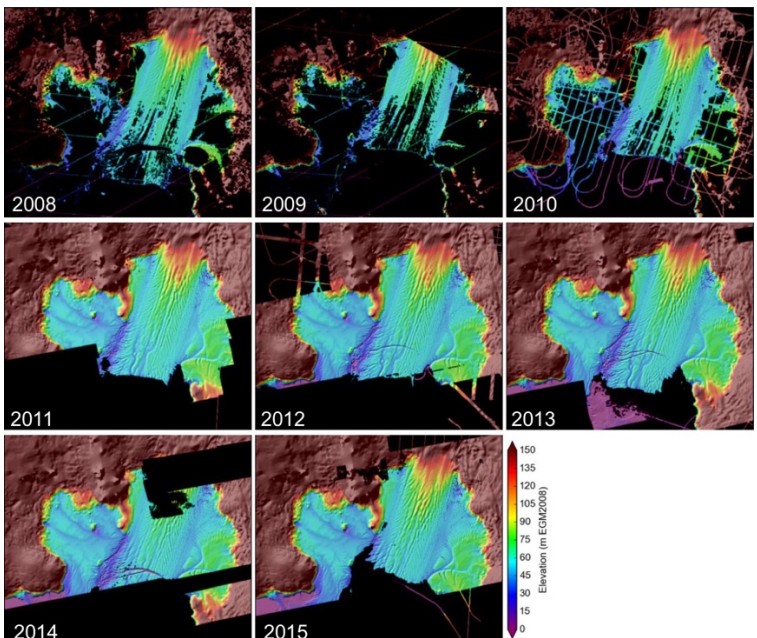

**Figure 6: Annual DEM composites using all available elevation data. Primary DEM sources are SPIRIT (top row), and WorldView/GeoEye (middle and bottom rows).**

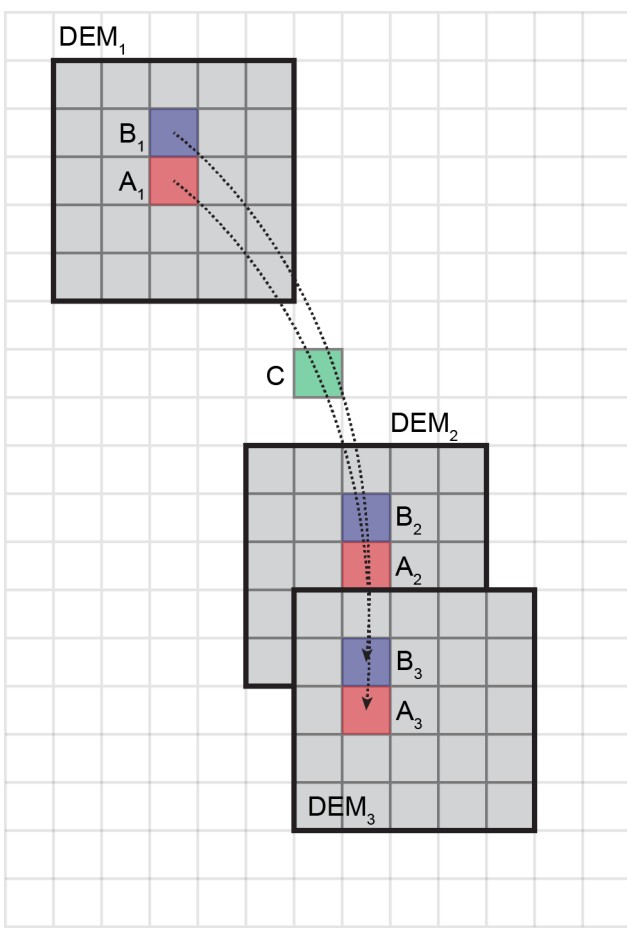

**Figure 7: Illustration of Lagrangian *Dh/Dt* calculation and basal melt rate distribution on a Eulerian grid (light gray). Three DEMs (medium gray) acquired at times $t_1$, $t_2$, and $t_3$ are resampled on this grid, with the same "features" A and B indicated as colored pixels. The position history for "particle" A is estimated using the velocity products described in Section 2.5, with paths indicated by dotted lines. Lagrangian *Dh/Dt* for A is calculated as $(h_{A2}-h_{A1})/(t_2-t_1)$. At each timestep along the path from $A_1$ to $A_2$ ($A_{12}$), we estimate *h* (from observed *Dh/Dt*), velocity divergence (from observed velocity time series), and the local flux divergence. Using Equations 10 and 12, the cumulative basal melt rate along the $A_{12}$ path is estimated. This procedure is repeated for "particle" B and all other "particles" in $DEM_1$ that intersect $DEM_2$. For the "along-flow distribution" approach, the cumulative basal melt rate for path $A_{12}$ is assigned to each Eulerian grid cell along path $A_{12}$, including grid cell C. This assignment is repeated for path $B_{12}$ and all other paths for $DEM_1$-$DEM_2$ particles, so that many basal melt rate values will be assigned to grid cell C. The median basal melt rate is calculated from all paths intersecting C. This median value at C (and all other grid cells with nonzero path count) are used to populate the along-flow distribution basal melt rate map for $DEM_1$-$DEM_2$. This process is repeated for $DEM_1$-$DEM_3$ and all other valid downstream $DEM_1$-$DEM_j$ combinations for the specified ~2-year time period. The same process is then repeated for all initial $DEM_i$, and full ice shelf composites are generated as described in Section 3.6.1. For the "initial-pixel" approach, the cumulative basal melt rate for path $A_{12}$ is assigned to cell $A_1$. This process is repeated for the basal melt rate along path $A_{13}$ and all other valid downstream $DEM_j$ to estimate "initial-pixel" stack median basal melt rate for $A_1$, and all other pixels in $DEM_1$. This "initial-pixel" stack median process is repeated for all valid $DEM_i$, and these products are combined to create full ice shelf composites as described in Section 3.6.2.**

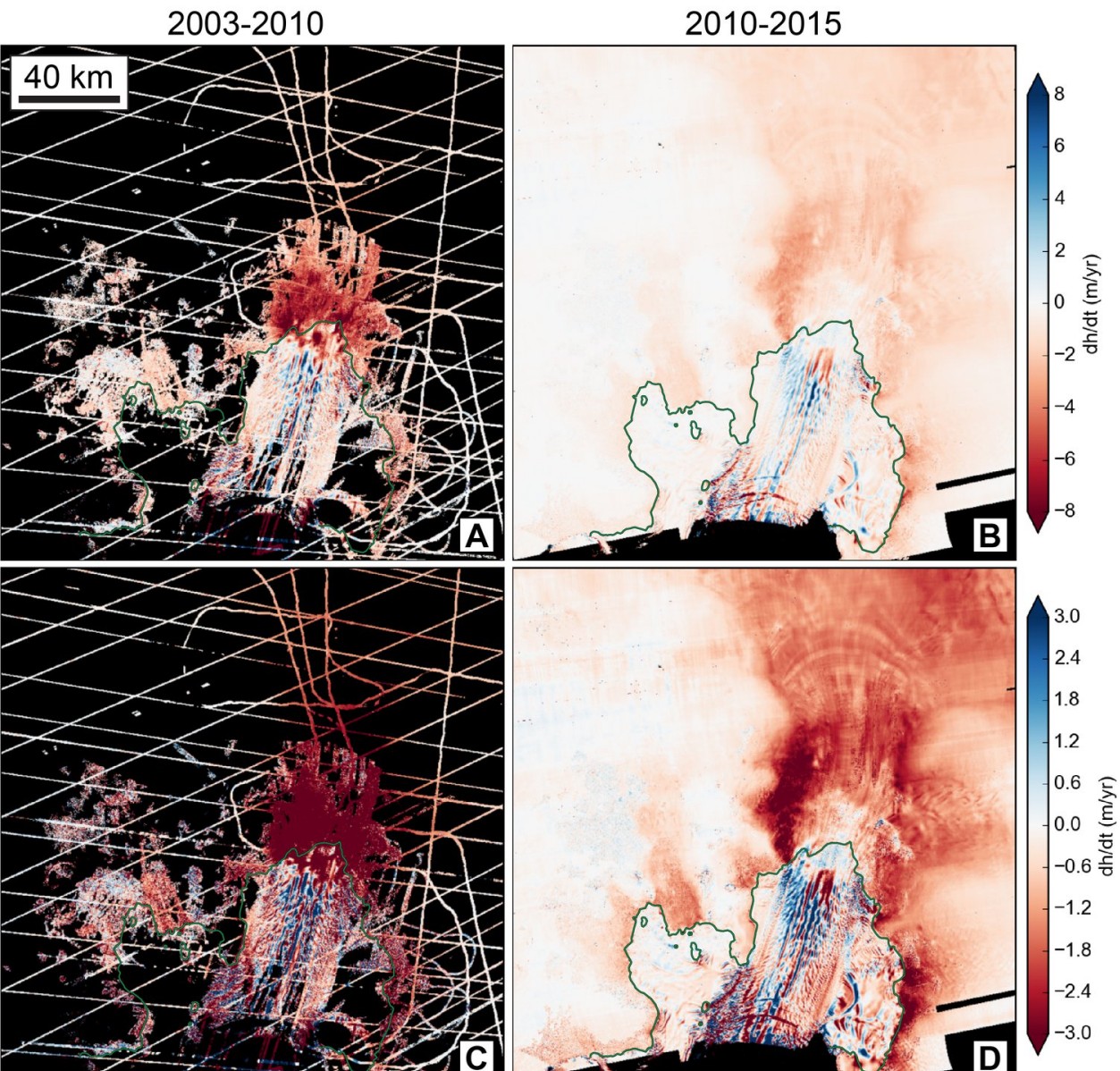

**Figure 8: Long-term Eulerian *dh/dt* trends for the PIG ice shelf and lower catchment. A) 2003-2010 *dh/dt* from ICESat, ATM/LVIS airborne altimetry and SPIRIT DEMs. B) 2010-2015 *dh/dt* from WorldView/GeoEye DEMs, SPIRIT DEMs and ATM/LVIS airborne altimetry. C+D) Same data as in A+B, but with enhanced contrast stretch to bring out details over main trunk.**

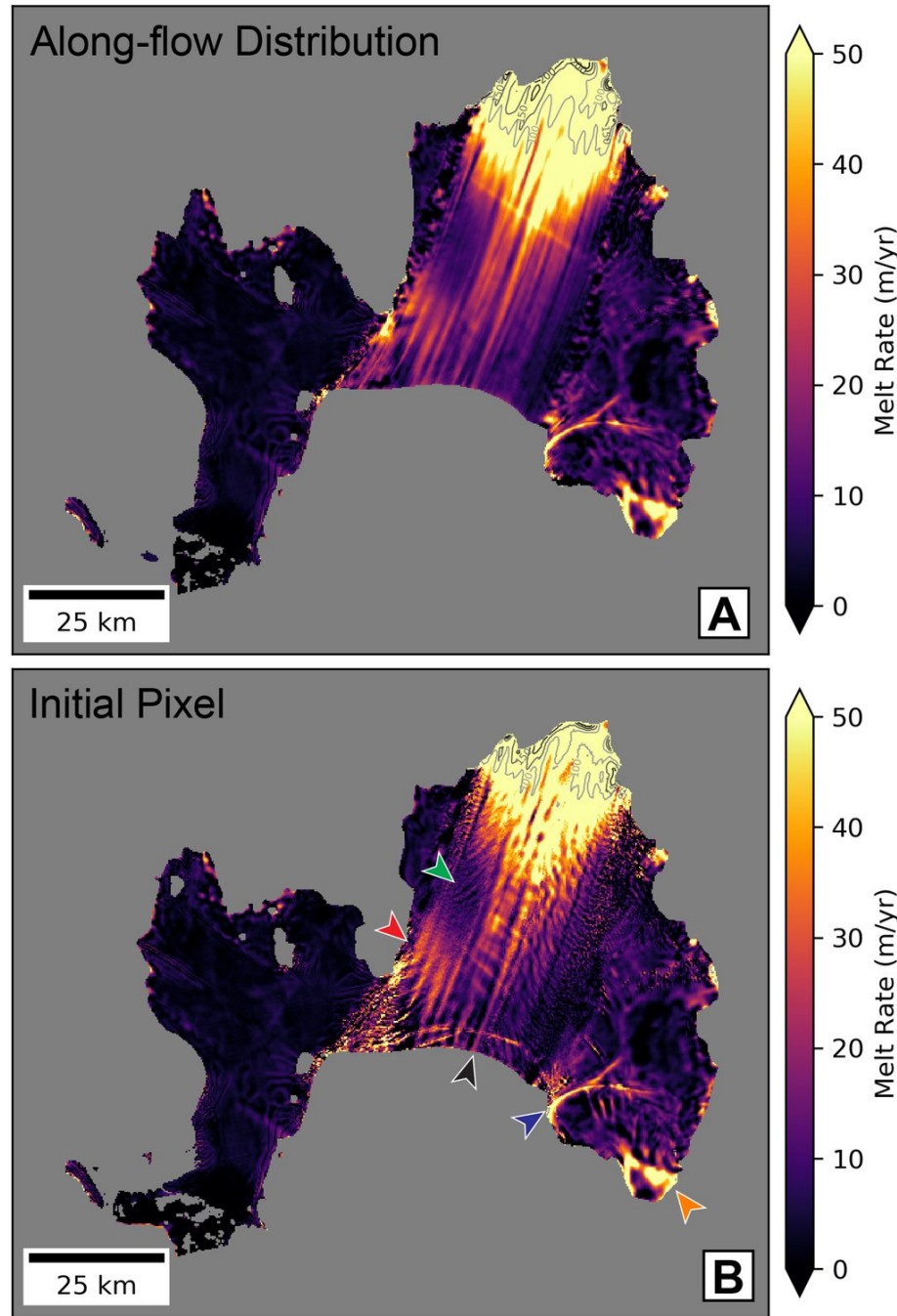

**Figure 9: Comparison of mean 2008-2015 basal melt rate composites using: A) 2-year "along-flow distribution" and B) 2-year "initial-pixel" methods. Color ramp shows 0-50 m/yr stretch for basal melt rates, with additional grayscale contours at 100, 150, 200 and 250 m/yr near the grounding line. The transverse features along the outer ice shelf centerline in B are related to enhanced melt within and near depressions/rifts (Shean et al., 2017). The transverse mid-ice-shelf artifact in A is the result of a seam artifact in one of the TerraSAR-X velocity mosaics. Colored arrows show features discussed in the text.**

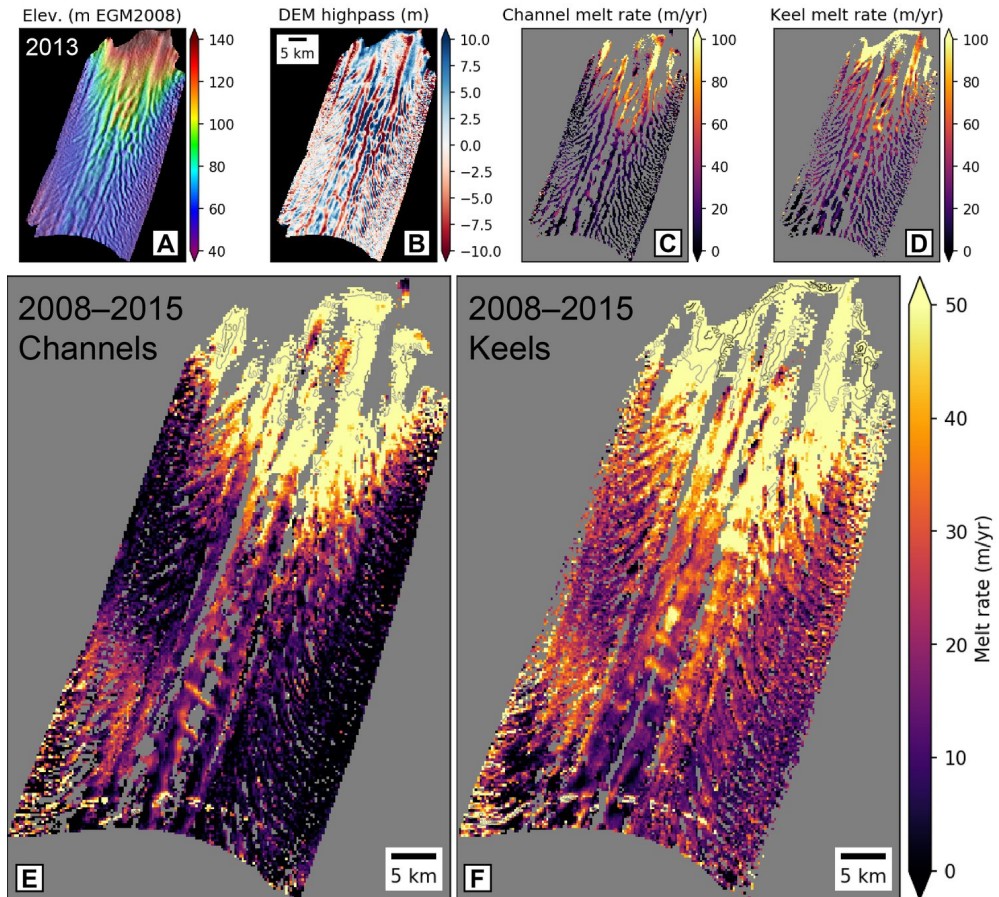

**Figure 10: Relationship between km-scale surface ridge/trough (basal keel/channel) features and initial-pixel basal melt rates for main ice shelf. Top row shows example products from one 2-year period (2013–2015): A) 256-m DEM mosaic, B) km-scale surface anomalies after high-pass filter (surface ridges are blue, surface troughs are red), C) basal melt rates for channels (where DEM anomaly is <-1 m), D) basal melt rates for keels (where DEM anomaly is >1 m). Note relatively high basal melt rates over longitudinal basal channels at distances of ~4-15 km from the grounding line in C. The bottom row shows channel (E) and keel (F) melt rate composites generated using all available 2-year products during the full 2008 to 2015 period. Color stretch of 0-50 m/yr highlights differences over the outer ice shelf, where higher basal melt rates are observed on keels. See Figure S5 for additional details.**

