# Peer review of "Ice shelf basal melt rates from a high-resolution DEM record for Pine Island Glacier, Antarctica"

_The Cryosphere, 2018_

## Referee Comment (RC1) · Anonymous Referee #1 · 19 Nov 2018

**GENERAL COMMENTS**

This paper describes the creation of maps of height change rates and basal melt rates for Pine Island Glacier's ice shelf, at high spatial resolution and with time-dependence. The authors use a wide range of input data products to produce maps that can resolve the structure of basal melting at scales of individual channel features. This will provide the community with valuable validation for increasingly high-resolution ocean models, and insights into critical processes determining the stability of PIG.

Most of the manuscript is a detailed technical description of the processes used to integrate the different data sets and optimize the output products in terms of basal melt rate. I'd hope to see more interpretation of the results in future papers. However, the manuscript is well written, a valuable description of how to get these important products, and even the limited "results" will be of significant interest, so I hope to see the paper published fairly quickly. While there are lots of comments, my major problem in reading the manuscript was in Section 2.9, where I needed more information to really understand the value of the initial-pixel approach to plotting the results. At the same time, this raised the question of whether there was complementary information available in Eulerian dh/dt processing, that might help with the problem that the Lagrangian processing over two years gives a large spatial along-flow average (a few km; larger than the channel and keel cross-flow scale); perhaps Eulerian is better near the grounding line, even with all the extra noise from lateral advection of surface topography?

SPECIFIC COMMENTS

Fig. 1: Add labels "Amundsen Sea" and "Pine Island Bay", and add the grounding line.

54-55: The laser and radar elevation data sets have very different issues with regard to their value for trend detection, and probably should be described separately.

Fig. 2: Would be useful to plot the most- and least-advanced grounding line on this figure, then explicitly refer to it when discussing GL migration.

82, Fig. 2c: the color scale for velocity on Fig. 2c saturates at 1 km/yr. But velocity is critical to the interpretation of the spatial scales of basal melt rates, so we need to see the fully resolved velocity field. At a minimum, include it as a full figure in the Supplement, with a color scale that resolves up to 4 km/yr even if it is stretched to improve resolution of speeds <1 km/yr as in Fig. 2c.

92, and Fig. 2: The location of the transverse seabed ridge is sufficiently important in this paper that it should be marked on at least one Main Text figure.

140-142: This is *very* technical, and won't make sense to anyone who hasn't worked on this already. The Shean (2016) thesis probably contains this information, but one option is to move this and other very technical stuff in the Supplement where you can

give it enough space without bringing the Main Text readers to a complete stop.

196: What is a "DEM-point time offset" ?

196-197: The accuracy of control points is small, and won't affect anything reported here. But (a) How is this accuracy determined? and (b) Is it really "accuracy", or "precision" of a specific measurement over the control?

199-200: I was confused by "with DEM-altimetry time offsets". Understand now (I think), but it needs more introductory text. I think the problem arises in part because you have these level-4 headers (2.2.1.1 and 2.2.1.2). If you write 2.2.1 as a single sub-section, and just use paragraph breaks to separate WV from SPIRIT discussion, then edit as a single section, it'll be clearer.

202-203, 210: I would like to see some sort of explanation for the 3.1 m bias". It is laser-to-visible, so there are no snow/firn penetration issues. If it could be traced to, say, different geoids, then it might be a spatially-dependent bias. Or, it is close to just making a sign error on MDT correction.

208-211: This is another complex "explanation" that requires more familiarity with the topic than many readers will have.

224: While it won't change your results, Andersen and Knudsen (2009) is a very old citation for MDT: if you are really using that old a product, you might want to change for more recent versions.

295-300: I think I follow what you are doing, but (a) you don't explain WHY the reverse-ordering is a good approach, and (b) it suggests to me that the optimum time stamp for the resulting product may not be as centered on the central image time as I would have expected.

346 ff: Everything before this point is describing the input data sets. Starting here, you are describing what you get in terms of products you really want to get at. Overall, the current Section 2 is too piecemeal, as evidenced by getting into level-4 subheaders.

Perhaps there is a better organization where Section 2 is just about the input data, then Section 3 (made up of 2.8-2.10) could be "Deriving elevation changes and basal melt rates". That way you'd never get to level-4 subheaders, and the transition from "data setup" to exploitation would be clearer.

362 through eq. (9): (a) "then compute ice-equivalent freeboard thickness: H = …". This slowed me down for a while. H is "ice thickness assuming no firn air", right? Maybe this would be clearer if eq. (8) was expressed "h = …" or it might just need better lead-in words before eq. (8). (b) I also had to think a bit about eq. (9). You apply divergence to firn air (d). However, you don't allow for any time rate of change of local firn air content. I think you end up accounting for this later, but here you describe it as "dropping the CONSTANT d from …" which seems unjustified at this time.

373 ff: If you decide that section 2.8 should stay in the existing section 2 (see comment two above here), then consider maybe Sections 2.9 and 2.10 into a separate section ("3. Basal melt rate estimation"?). Whatever you do, restructure to avoid level-4 sub-headers.

405-410: This para needs to be revised. Start by explaining the concept of channels and keels. At the moment, you introduce the idea of "vertically variable" melt rate before I have any idea in what way a melt rate might vary "vertically" at a single site.

444-450, and more generally 419-451: I found this text very difficult to follow the first few times I read it, and maybe it is still a problem. Given that Lagrangian-derived melt rate is only calculated on 1-2 year time periods, how can you really get information at smaller scales than is associated with advection over this time? The initial-pixel product is more detailed (Fig. 8), but you don't really know anything about whether it varies on such small scales. The description of the differences and limitations of the two methods is honest, so it's okay as-is, but it raises the question of whether there is valuable complementary information in an EULERIAN dh/dt calculation, especially with regard to the region near the grounding line where melt rates change rapidly. Maybe

the extra noise in the Eulerian calculation is worth it in this case?

486: The velocity error derived from look angle and elevation uncertainty makes sense. However, you cited tide and IBE error as 0.1 m. Why do you need to use the full tide, rather than the error, in this calculation?

Figure 7: (a) Add grounding line(s) to these plots. (b) Maybe make the ICESat and ATM tracks a bit wider so it's easier to see the color shading of the tracks. (c) {In caption}: There's no such thing as "ICESat-1" is there? Just "ICESat". (d) A "North" arrow would be useful (on one panel).

Figure 8: I found the arrows harder to find than they should be. Perhaps use a white outline around the edge of the arrow, and/or add a shaft to the arrowhead, and/or use lighter shades of each color as the background is generally dark.

537-540: Even with the colorbar stretch in the lower panels of Fig. S4, it is hard to see detail of melt rate on the North ice shelf, and even much of the South. Maybe choose an even smaller range for the lower panels of Fig. S4, or even add a third set of panels at a lower range, designed specifically to highlight North and South ice shelves.

609-610: How can Payne et al. (2007) be evidence both for "significantly higher than past estimates" but also "more consistent with the Payne et al. (2007) estimates" ?

611-613: You can't really claim that your estimates are less than past estimates, when they "fall within reported uncertainty". It would be better to acknowledge first that they are the same within uncertainty, but then speculate on "maybe they really are smaller, which we'd explain as . . ."

625-628: The other advantage to Ross and FRIS is that theya re further south where ICESat orbits are closer together.

TECHNICAL CORRECTIONS

Given the oceanographic interest, the use of "shelf" instead of "ice shelf" can be confusing. I'd recommend always saying "ice shelf".

When referring to a paper, I tend to go with "BY Smith et al." rather than "IN Smith et al." unless it refers to a specific items like a figure or an equation. Just a more direct credit to the authors.

The use of the dash between dates is variable and confusing. "from 2008-2015" might mean "from 2008 to 2015" or "during the period 2008-2015". Try to read it out, and see if just using "to" or "between XXXX and XXXX" would work better.

Present/past tense: Wanders a bit, e.g., line 168, probably should have "mask" => "masked" and "remove" => "removed". Also in Section 2.1.3, and several other places.

Try to avoid "Figure/Table X shows that". Usually it's possible to give the science, then the figure cite parenthetically.

556, 603, 630: Remove all "We note that"; not needed.

"ICESat-1": My understanding is that,e ven though there is now an ICESat-2, the original laser mapping mission is still called "ICESat", not "ICESat-1".

10: "directly and indirectly" presumably refers to direct loss from the ice SHELF, and indirect loss from the ice SHEET, but it immediately slows down the reader.

38-48: Break this sentence up; use separate sentences for each major methodology.

74-75: I expected to see a cite to Shepherd et al. 2018 (IMBIE) here.

82: "Two additional ice shelves". Not really. North and South ice shelves are part of PIG, and you count them as part of PIG.

83-86: Confusing construction, and I'm not sure it's even true. It is true that velocities of the N and S ice shelves are relatively small, but the quoted thickness ranges are similar to most of the main trunk. In fact, the S shelf is thicker, on average, than most of the Main trunk. (Fig. 2d treated as a thickness proxy).

87, Fig. 2A: Please add the catchment boundaries to this figure.

91-92: Probably don't need a paragraph break here.

109-110: Sentence starting "Water at . . ." needs a cite or two.

135-136: Fig. 1 shows the "ASE" coastline, not the "West Antarctic" coastline.

186: maybe "extend the . . . record BACK to 2003". When I first read it, sounded like 2003 was later than most data.

201: revise to "There is a . . . (Fig. 3, Table 2)."

242-243: Isn't there also a Ben Smith (co-author) explanation of this approach for processing of ICESat along-track data?

248: "the main shelf and fast-flowing trunk". Aren't these the same thing? Okay, no, I guess the "trunk" is the grounded ice, so say "main {ice} shelf and fast-flowing GROUNDED trunk"

263-264: probably just use "high-priority areas such as . . ." and delete the "i.e., and out parentheses.

312: "m/px" is a strange abbreviation. Maybe cleaner to use "m/pixel"

314: "over a ∼11-189 day period" reads awkwardly.

317: I have no idea what "barycentric interpolation" is.

336: {FYI} Mueller et al. (2012; JGR) used bed interpolation along flow lines to make a sub-ice-shelf bathymetry grid for Larsen C.

379: "Lagrangian Dh/dt basal melt rates" doesn't make sense. Maybe "basal melt rates based on Lagrangian height-change measurements"

399: "from A single, fixed"

400: "step but, in practice, . . ." (move first comma).

401: "useD", "substituteD"

521: "downstream of THE grounding line"

555: "over keels" is ambiguous; "on keels" or something else would be clearer.

577: Why quotation marks around "resolution" ?

———————————————

---

## Referee Comment (RC2) · Anonymous Referee #2 · 25 Mar 2019

This paper presents an extensive study of high-resolution satellite DEMs for determining elevation changes and ice-shelf basal melting of Pine Island Glacier in unprecedented detail. The amount of data and complexity of the analysis are impressive and well justified. A lot of care has been taken to find optimum methodologies and complementary ways of representing basal melt rates in a spatial domain that is constantly evolving. The high technical quality of the paper will make it a benchmark for related studies in the future, and the detailed results of basal melting will be highly useful for modellers.

The glaciological part of the paper is somewhat drowning in all the technicalities, but has a high potential for follow-up papers concerning links with ice-ocean interactions. It is therefore still well suited for a journal like the Cryosphere. The main things I would

like to see improved before publication are the overly fragmented sections and the figures which are sometimes too excessive and not as informative as they could have been. More details on that are provided below, first on the figures and then on the manuscript text in line-by-line comments and edits which are mostly of minor nature.

Figure comments and suggestions:

Fig. 1: Although an impressive data set, I do not see the purpose of these regional DEM figures when the paper is all about PIG. Figure 2d would be a better opening figure, potentially with Fig. 1A or similar as an inset to show the location of PIG within West Antarctica. The present Fig. 1 could still be included in the Supplement, and in that case grounded ice and ocean areas could be masked out in panel b to highlight the ice shelves.

Fig. 2: The upper three panels could be split into a separate figure to better highlight the detailed and nice ice shelf one. The ice shelf outline is hardly visible in panels b and c, which should be improved for context, preferentially with bedrock and velocity close-ups of the ice shelves like the upper third panel of Fig. S1. That would be a good opportunity to highlight the "transverse seabed ridge" in the main paper, as well as velocity variations on the main ice shelf which are totally saturated in Fig. 2c. In that case, a new multi-panel figure would definitely be needed and could replace the current Fig. 1.

Fig. 2 caption: I assume that it is over Bedmap2 surface hillshade, not "Bedmap2 DEM". For panel B; "...anisotropic interpolation of available data" (or be more specific). Unless grids or coordinates are included in any of the figures, the caption should say something about the location and map projection.

Fig. 3: Could be moved to the supplement, but since this paper focuses so much on methodology it is okay to keep here. Please define in the caption what the dashed lines (mean or median of all errors?) and error bars represent (16-84% spread?). The small text at the top is somewhat cryptic and should be removed or included in the caption.

[Figure]

Fig. 4: This is a powerful illustration of the importance of careful co-registration for studies like this. To be improved: Make panel numbers as in preceding figures. Define what "Count" is. Include the PIG grounding line in white since that partly coincides with where the DEM variability is higher.

Fig. 5: Panel numbers missing.

Fig. 6: This figure and the elaborate caption give a good explanation for a very complex methodology, but repeats and overlaps with the manuscript text which is unnecessarily fragmented into a series of sub-sections. These parts need to be better harmonized, potentially by moving much of the caption into the main text and removing repeated information from there.

Fig. 7: A shaded relief would be a nicer background than black in panels a and c. Ice shelf outlines (in white) would also be helpful in all panels to see relations with dh/dt patterns.

Fig. 8: Switch order of panels similar to the order of discussion in the text? I am not sure if the color scale is optimum, a lot of the ice shelf appear dark with no visual contrast.

Fig. 9: It is a very good idea to separate channels and keels in the figure, but in the end it is actually difficult to see clear differences between panels and that also makes some of the main conclusions of the paper weaker. Have you tried other color bars or different scaling to see if differences may come out more clearly? Panel numbers are also missing.

Fig. S1: Include panel numbers. The header "anisotropic interpolation" is confusing unless more details are provided (interpolation of what?).

Fig. S2: Something wrong with the reference here. I assume you want to refer to the similar Fig. 3 in the main manuscript. See also my comment to that figure.

Fig. S4: See comment to Fig. 8. Although different colorbar stretches here are helpful
for interpretation of Fig. 8, I still feel that a lot of area appear dark without visible contrast.

Line-by-line comments and edits:

62-66: All the information and parentheses make this sentence hard to read. Split/rewrite or remove some content. This is also good to keep in mind elsewhere in the manuscript during revisions.

67: To be clear; "... across the grounding line of the South ice shelf"

82: Fig. 2C is not a good reference here since it saturates at 1 km/yr. Remove.

83: The "South shelf" has already been introduced at L67, but more clearly here. Perhaps the order of the text should be switched or L67 can be simplified/removed.

92: This is an important feature and therefore it is somewhat unfortunate that the reader is only referred to the supplement. However, it is also visible in Fig. 2B if the ice shelf outline is made thicker to highlight the relevant area, i.e. reference (Fig. 2B and S1).

198: Four levels of sections is quite elaborate for a single research paper. I think that subdivision into 2.2.2.1 and 2.2.1.2 is more confusing than helpful.

228: I assume that this applies to the ice-shelf side of the grounding line. Suggestion for rewrite: "To provide a smooth transition from fully grounded ice to freely floating ice, we defined the coefficient ... downstream of the grounding line:"

Eq. 3: $t$ is not defined

257-259: It is not clear what the "tilt tolerance" is and how it was used.

267: Define or remove CE90/LE90

335: Mention "anisotropic" here to make the link with Fig. S1 obvious.

345: How was the finer interpolation of the coarse RACMO data done?

352: How was firn air content removed? As a constant or time-variable?

353: Here you talk about surface accumulation, earlier SMB. Be consistent.

365: What do you mean by "dropping the constant d"? It is probably reasonable to treat it as a constant, but that should be briefly discussed/justified, e.g. in section 2.7.

418: I think the word "product" is misleading here since it's more a description of how the calculation is done, whereas composites/mosaics are described later.

443: Is a separate section needed? It is all almost about the initial-pixel approach, and the last sentence can be moved to line 417.

482: The simplified assumption of a constant firn air content in space and time is probably a minor issue for PIG, but not for most other ice shelves in Antarctica. A brief discussion or recapture of the cited reference would therefore be appropriate.

559: In this section you discuss characteristics of channel and keel melting shown in Fig. 9, but refer to arrows in Fig. 8. That is confusing, and even more so when some features like the "prominent longitudinal keel" is generally visible in Fig. 8, but not in Fig. 9 where keels and channels are separated.

593: It is very hard to actually see this in Fig. 9. Is it possible to find a better color scheme that makes this key point more convincing?

656: FDM is not used in the manuscript as far as I can see (constant firn air content)

Table 3: Define "n"

[Figure]

---

## Author Comment (AC1) · 17 Jul 2019

**GENERAL COMMENTS**

This paper describes the creation of maps of height change rates and basal melt rates for Pine Island Glacier's ice shelf, at high spatial resolution and with time-dependence. The authors use a wide range of input data products to produce maps that can resolve the structure of basal melting at scales of individual channel features. This will provide the community with valuable validation for increasingly high-resolution ocean models, and insights into critical processes determining the stability of PIG.
*We thank the reviewer for a positive assessment.*

Most of the manuscript is a detailed technical description of the processes used to integrate the different data sets and optimize the output products in terms of basal melt rate. I'd hope to see more interpretation of the results in future papers. However, the manuscript is well written, a valuable description of how to get these important products, and even the limited "results" will be of significant interest, so I hope to see the paper published fairly quickly.
*Yes, this paper was intended to present the methodological details and shelf-wide melt rates. A follow-on paper describes the spatiotemporal variability with more interpretation.*

While there are lots of comments, my major problem in reading the manuscript was in Section 2.9, where I needed more information to really understand the value of the initial-pixel approach to plotting the results.
*We added additional clarification in Section 2.9 to emphasize the importance of analyzing local elevation change with initial-pixel approach.*

At the same time, this raised the question of whether there was complementary information available in Eulerian dh/dt processing, that might help with the problem that the Lagrangian processing over two years gives a large spatial along-flow average (a few km; larger than the channel and keel cross-flow scale); perhaps Eulerian is better near the grounding line, even with all the extra noise from lateral advection of surface topography?
*Yes, there is valuable information in the Eulerian dh/dt products. This is obvious over grounded ice. As the reviewer points out, there is a transition zone near the grounding line, where this information could be complementary to the Lagrangian Dh/Dt products over floating ice. We consider the evolution of 2-year Eulerian dh/dt with the 2-year Lagrangian Dh/Dt melt rates in the follow-on manuscript describing temporal changes in the basal melt rates.*
*While it is beyond the scope of this paper, a more complex function could potentially be used to merge the Lagrangian Dh/Dt and Eulerian dh/dt values near the grounding zone.*
*We provide additional comments on this subject under the "Specific Comments" below.*

**SPECIFIC COMMENTS**

Fig. 1: Add labels "Amundsen Sea" and "Pine Island Bay", and add the grounding line.
*We added "Amundsen Sea" label to Figure 1. Did not add "Pine Island Bay" as it would clutter the figure, and did not add grounding line, as this is available for PIG in subsequent figures.*

54-55: The laser and radar elevation data sets have very different issues with regard to their value for trend detection, and probably should be described separately.
*While this is true, we feel that our existing language is generic enough to be applied to both. Modified slightly to "Elevation data from satellite laser and radar altimetry are further limited by large footprints and sparse repeat-track spacing, with increased uncertainty over areas with non-negligible slopes and/or roughness."*

Fig. 2: Would be useful to plot the most- and least-advanced grounding line on this figure, then explicitly refer to it when discussing GL migration.
*Most and least advanced over what time period, and presumably Fig 2D? We feel that plotting multiple grounding lines could make the figure too complicated, and obscure details over the shelf that are important for subsequent discussion. The (Joughin et al., 2016) paper should offer sufficient context for grounding line evolution.*

82, Fig. 2c: the color scale for velocity on Fig. 2c saturates at 1 km/yr. But velocity is critical to the interpretation of the spatial scales of basal melt rates, so we need to see the fully resolved velocity field. At a minimum, include it as a full figure in the Supplement, with a color scale that resolves up to 4 km/yr even if it is stretched to improve resolution of speeds <1 km/yr as in Fig. 2c.
*The goal with Figure 2C was to show catchment-wide velocities and bring out detail in the south shelf tributary. We elected not to use a logarithmic color ramp, which required saturating the linear color ramp. Figure 1 in (Shean et al., 2017) TC uses a 0-4 km/yr color ramp for the shelf with the same grounding line, and we added a note referring readers to this figure.*

92, and Fig. 2: The location of the transverse seabed ridge is sufficiently important in this paper that it should be marked on at least one Main Text figure.
*Both reviewers commented on this. We added a label to Figure 2B. This feature is labeled in Figure S1, and we added a note to Figure 2 caption referring the reader to Figure S1 for detailed bathymetry.*

140-142: This is *very* technical, and won't make sense to anyone who hasn't worked on this already. The Shean (2016) thesis probably contains this information, but one option is to move this and other very technical stuff in the Supplement where you can give it enough space without bringing the Main Text readers to a complete stop.
*We agree that this is technical, but it is one sentence. Additional details are in the ISPRS paper, (not the thesis), which is what is cited, but the specific settings for PIG DEM generation are*

*different. As this is largely a methods paper, we feel it is important to include this note for improved reproducibility, and we modified language slightly to indicate the settings are intended for advanced ASP users.*

196: What is a "DEM-point time offset" ?
*This was described in the preceding sentence. We added a new sentence to clarify "We removed points with time offset between the point and DEM timestamp (|t_altimetry - t_DEM |) of >1 year." and removed the portion of the sentence that caused confusion.*

196-197: The accuracy of control points is small, and won't affect anything reported here. But (a) How is this accuracy determined? and (b) Is it really "accuracy", or "precision" of a specific measurement over the control?
*This estimate is for absolute accuracy of ICESat, ATM and LVIS. We added a note referring the reader to (Shean et al., 2016) ISPRS paper, section 5.1, which discusses this issue.*

199-200: I was confused by "with DEM-altimetry time offsets". Understand now (I think), but it needs more introductory text. I think the problem arises in part because you have these level-4 headers (2.2.1.1 and 2.2.1.2). If you write 2.2.1 as a single sub-section, and just use paragraph breaks to separate WV from SPIRIT discussion, then edit as a single section, it'll be clearer.
*Good suggestion, we removed the level-4 headers, and used the newly defined "|t_altimetry - t_DEM|" instead of "DEM-altimetry time offset".*

202-203, 210: I would like to see some sort of explanation for the 3.1 m bias". It is laser-to-visible, so there are no snow/firn penetration issues. If it could be traced to, say, different geoids, then it might be a spatially-dependent bias. Or, it is close to just making a sign error on MDT correction.
*(Shean et al. 2016) section 6.1.1 includes a discussion of this bias and potential causes. More recent versions of ASP include an atmospheric refraction correction that should mitigate some of this bias. We added "Uncorrected DEMs had an initial mean vertical bias of +3.1 m above the altimetry data (Figure 3), as discussed in section 6.1.1 of Shean et al. (2016), and we applied a -3.1 m vertical correction…"*

208-211: This is another complex "explanation" that requires more familiarity with the topic than many readers will have.
*This is a note on additional processing required for one problematic SPIRIT DEM, and we feel it is important to document for reproducibility.*

224: While it won't change your results, Andersen and Knudsen (2009) is a very old citation for MDT: if you are really using that old a product, you might want to change for more recent versions.
*We agree that this will not change the results. We welcome the reviewer to suggest an updated reference to use here.*

295-300: I think I follow what you are doing, but (a) you don't explain WHY the reverse- ordering is a good approach, and (b) it suggests to me that the optimum time stamp for the resulting product may not be as centered on the central image time as I would have expected.

*The paragraph above this section and introductory sentence provide justification for why this ordering is necessary. We also tested ordering with earliest timestamps on top, and timestamps closest to an arbitrary date (e.g., Jan 1) on top. In the end, the reverse ordering scheme provided best continuity for the available DEMs. Separate timestamp layers provide the corresponding DEM timestamp at each pixel in the ordered mosaics, for later analysis.*
*As described in the text, we also generated weighted average composites for all available DEMs each season. None of this is relevant for the basal melt rate calculations, which use individual DEMs, not these mosaics.*

346 ff: Everything before this point is describing the input data sets. Starting here, you are describing what you get in terms of products you really want to get at. Overall, the current Section 2 is too piecemeal, as evidenced by getting into level-4 subheaders.
Perhaps there is a better organization where Section 2 is just about the input data, then Section 3 (made up of 2.8-2.10) could be "Deriving elevation changes and basal melt rates". That way you'd never get to level-4 subheaders, and the transition from "data setup" to exploitation would be clearer.

*This is a great suggestion, and helps with organization. We separated into Section 2 and Section 3 as suggested.*

362 through eq. (9): (a) "then compute ice-equivalent freeboard thickness: H = ...". This slowed me down for a while. H is "ice thickness assuming no firn air", right? Maybe this would be clearer if eq. (8) was expressed "h = ..." or it might just need better lead-in words before eq. (8). (b) I also had to think a bit about eq. (9). You apply divergence to firn air (d). However, you don't allow for any time rate of change of local firn air content. I think you end up accounting for this later, but here you describe it as "dropping the CONSTANT d from ..." which seems unjustified at this time.

*To avoid confusion removed "(after removing firn-air content d)" from the sentence above eq 4, so that this reads "ice-equivalent thickness H can be expressed as... ". We also modified eq 8 to use Hf instead of H.*
*Yes, we are assuming time-invariant (constant) firn air content. We modified the text above Equation 9 to clarify: "This ice-equivalent freeboard thickness Hf can then be substituted for H into in Equation 7. We assume that any changes in d, $\rho\_w$ and $\rho\_i$ are negligible during our study period, and the DHf/Dt term reduces to Lagrangian surface elevation change (Dh/Dt), resulting in a modified mass-conservation expression for a column of floating ice:"*

373 ff: If you decide that section 2.8 should stay in the existing section 2 (see comment two above here), then consider maybe Sections 2.9 and 2.10 into a separate section ("3. Basal melt rate estimation"?). Whatever you do, restructure to avoid level-4 sub- headers.
*We restructured as suggested in previous comment*

405-410: This para needs to be revised. Start by explaining the concept of channels and keels. At the moment, you introduce the idea of "vertically variable" melt rate before I have any idea in what way a melt rate might vary "vertically" at a single site.

*We modified this paragraph, but do not feel it is necessary to explain the basic concept of channels and keels, as these are introduced earlier in the text. The existing description as "features with variable draft" should be sufficient. Changed to "End member #1 assumes a fixed 3D "melt rate field" within the ocean cavity beneath the PIG ice shelf that varies spatially but not temporally, so that features with variable draft (i.e., keels and channels) melt at different rates as they advect through this field."*

444-450, and more generally 419-451: I found this text very difficult to follow the first few times I read it, and maybe it is still a problem.

*We acknowledge that this is a complex concept. This section underwent several revisions before submission to improve readability, and we feel it is sufficient. If the reviewer has specific comments on how to improve, we are open to make additional changes.*

Given that Lagrangian-derived melt rate is only calculated on 1-2 year time periods, how can you really get information at smaller scales than is associated with advection over this time? The initial-pixel product is more detailed (Fig. 8), but you don't really know anything about whether it varies on such small scales.

*We disagree with this conclusion. This is precisely why we outlined the two end members for melt rate field and explored two different approaches for melt rate calculation.*
*The "along-flow" approach can be thought of as a form of tomography - multiple features with different start and end points pass through the same location on an Eulerian grid can provide enhanced detail on the melt rate field (end member #1) at that location. The "initial-pixel" approach is calculated at the inherent resolution of the DEMs, and if melt rates are dictated by local geometry (end member #2), then we are resolving melt rates at these scales.*

The description of the differences and limitations of the two methods is honest, so it's okay as-is, but it raises the question of whether there is valuable complementary information in an EULERIAN dh/dt calculation, especially with regard to the region near the grounding line where melt rates change rapidly. Maybe the extra noise in the Eulerian calculation is worth it in this case?

*This is a good question. One can consult Figure 8 (long-term Eulerian dh/dt, now with grounding line plotted for context) for some insight. After the ice plain ungrounding events that ended in ~2008, there is limited elevation change (even slightly positive) immediately downstream of the new groundling line. However, we continue to see significant thinning in the Lagrangian Dh/Dt products due to basal melting and divergence. Compare Figure 8D with Figure 9 (updated numbering scheme).*
*As mentioned in general comments, we consider this question further in the follow-on manuscript.*

486: The velocity error derived from look angle and elevation uncertainty makes sense. However, you cited tide and IBE error as 0.1 m. Why do you need to use the full tide, rather than the error, in this calculation?

*The velocity maps were generated using the established methodology of Joughin (2002), which does not correct for tidal variations over floating ice shelves, so the 0.5 m accounts for expected tidal magnitude. As described in section 2.2.2, we corrected surface elevation data over floating portions of the PIG ice shelf for tide and the inverse barometer effect. The quality of these corrections was evaluated in (Shean et al. 2017), which is cited in the text.*

Figure 7: (a) Add grounding line(s) to these plots. (b) Maybe make the ICESat and ATM tracks a bit wider so it's easier to see the color shading of the tracks. (c) {In caption}: There's no such thing as "ICESat-1" is there? Just "ICESat". (d) A "North" arrow would be useful (on one panel).

*We added the grounding line, which improves interpretation.*
*The tracks are part of the gridded dataset, so their width is based on the grid cell size. We prefer to keep them sparse, which is closer to the actual ICESat footprint.*
*Good catch - yes, it's ICESat, not ICESat-1 - fixed all instances.*

Figure 8: I found the arrows harder to find than they should be. Perhaps use a white outline around the edge of the arrow, and/or add a shaft to the arrowhead, and/or use lighter shades of each color as the background is generally dark.

*Added white outline to arrows in Figure 8.*

537-540: Even with the colorbar stretch in the lower panels of Fig. S4, it is hard to see detail of melt rate on the North ice shelf, and even much of the South. Maybe choose an even smaller range for the lower panels of Fig. S4, or even add a third set of panels at a lower range, designed specifically to highlight North and South ice shelves.

*We feel the perceptually uniform color ramp in Fig S4 provides sufficient contrast to distinguish meaningful melt rate variations on the North and South shelves. Melt rate differences of <5 m/yr are within uncertainty.*

609-610: How can Payne et al. (2007) be evidence both for "significantly higher than past estimates" but also "more consistent with the Payne et al. (2007) estimates" ?

*Good catch. Removed citation from first sentence.*

611-613: You can't really claim that your estimates are less than past estimates, when they "fall within reported uncertainty". It would be better to acknowledge first that they are the same within uncertainty, but then speculate on "maybe they really are smaller, which we'd explain as ..."

*This is true. We modified the text to clarify.*

625-628: The other advantage to Ross and FRIS is that theya re further south where ICESat orbits are closer together.

*Good point. Added a note on this.*

**TECHNICAL CORRECTIONS**

Given the oceanographic interest, the use of "shelf" instead of "ice shelf" can be confusing. I'd recommend always saying "ice shelf".

*This is a good suggestion. We attempted to correct all ambiguous instances of "shelf" to "ice shelf"*

When referring to a paper, I tend to go with "BY Smith et al." rather than "IN Smith et al." unless it refers to a specific items like a figure or an equation. Just a more direct credit to the authors.

*This is also a good suggestion. Modified where appropriate.*

The use of the dash between dates is variable and confusing. "from 2008-2015" might mean "from 2008 to 2015" or "during the period 2008-2015". Try to read it out, and see if just using "to" or "between XXXX and XXXX" would work better.

*We attempted to modify instances where this might be confusing. We attempted to consistently use em-dash for date ranges. Defer to TC editorial staff on best practices.*

Present/past tense: Wanders a bit, e.g., line 168, probably should have "mask" => "masked" and "remove" => "removed". Also in Section 2.1.3, and several other places.

*Fixed these instances, and attempted to improve consistency throughout paper.*

Try to avoid "Figure/Table X shows that". Usually it's possible to give the science, then the figure cite parenthetically.

*We appreciate the suggestion, but this is an author-specific style decision.*

556, 603, 630: Remove all "We note that"; not needed.

*Good point. Removed all instances*

"ICESat-1": My understanding is that,e ven though there is now an ICESat-2, the orig- inal laser mapping mission is still called "ICESat", not "ICESat-1".

*Fixed as described in previous response.*

10: "directly and indirectly" presumably refers to direct loss from the ice SHELF, and indirect loss from the ice SHEET, but it immediately slows down the reader.

*Yes, this refers to the loss from sub-ice-shelf melting and the dynamic response leading to increased discharge. But removed to avoid confusion*

38-48: Break this sentence up; use separate sentences for each major methodology.

*We disagree that separate sentences are needed for each methodology and prefer the concise presentation (albeit with many references).*

74-75: I expected to see a cite to Shepherd et al. 2018 (IMBIE) here.

*Added*

82: "Two additional ice shelves". Not really. North and South ice shelves are part of PIG, and you count them as part of PIG.
*Good point. We modified to read:*
*"Shear margins with width ~2-4 km separate the main shelf from the northeast ("North shelf") and southwest ("South shelf") sectors of the PIG ice shelf (Figure 2D)"*

83-86: Confusing construction, and I'm not sure it's even true. It is true that velocities of the N and S ice shelves are relatively small, but the quoted thickness ranges are similar to most of the main trunk. In fact, the S shelf is thicker, on average, than most of the Main trunk. (Fig. 2d treated as a thickness proxy).
*This is a valid point. We were comparing to the thickness closer to the main shelf grounding line. We removed discussion of thickness for N and S shelves, and only discuss velocity.*
*Changed to:*
*"In general, surface velocity is relatively slow (<100–500 m/yr) over the North and South shelves, except for a fast-flowing tributary of the South ice shelf with velocity of ~1 km/yr and thickness of ~1 km near the grounding line (Figure 2)"*

87, Fig. 2A: Please add the catchment boundaries to this figure.
*We do not feel this is necessary, as the topography in 2A shows the catchment, and it is not relevant for the main focus of the paper.*

91-92: Probably don't need a paragraph break here.
*Removed*

109-110: Sentence starting "Water at ..." needs a cite or two.
*Added (De Rydt et al., 2014; Dutrieux et al., 2014b) citations to the text.*

135-136: Fig. 1 shows the "ASE" coastline, not the "West Antarctic" coastline.
*This is a fair point. But it also spans the Bellinghausen Sea, and perhaps a portion of the Ross Sea, depending on where partitions are drawn.*
*Changed to "for the Amundsen and Bellinghausen Sea coastline of West Antarctica"*

186: maybe "extend the ... record BACK to 2003". When I first read it, sounded like 2003 was later than most data.
*Changed*

201: revise to "There is a ... (Fig. 3, Table 2)."
*Changed to:*
*"The ICP co-registration provided translation corrections for 368 of 575 DEMs over the PIG catchment, with a significant improvement in multiple quality metrics following co-registration (Figure 3, Table 2)."*

242-243: Isn't there also a Ben Smith (co-author) explanation of this approach for processing of ICESat along-track data?

*We are not entirely clear about the reviewer's suggestion here - is there a citation we should consider including?  D. Shean implemented the methodology described in the text after discussion with B. Smith.*

248: "the main shelf and fast-flowing trunk". Aren't these the same thing? Okay, no, I guess the "trunk" is the grounded ice, so say "main {ice} shelf and fast-flowing GROUNDED trunk"

*Changed to "main ice shelf and fast-flowing grounded ice stream"*

263-264: probably just use "high-priority areas such as ..." and delete the "i.e., and out parentheses.

*Changed as suggested*

312: "m/px" is a strange abbreviation. Maybe cleaner to use "m/pixel"

*This is a common abbreviation in the remote sensing literature.  Changed to "...500-m ALOS and LS8 products…"*

314: "over a ~11-189 day period" reads awkwardly.

*Deleted this sentence, as details are in citations.*

317: I have no idea what "barycentric interpolation" is.

*https://codeplea.com/triangular-interpolation*

336: {FYI} Mueller et al. (2012; JGR) used bed interpolation along flow lines to make a sub-ice-shelf bathymetry grid for Larsen C.

*We were not aware of this paper, thank you for the reference.  We added a citation, though the interpolation methodology used here is continuous for the bed and bathymetry, with many more observational constraints and more sophisticated regularization.*

379: "Lagrangian Dh/dt basal melt rates" doesn't make sense. Maybe "basal melt rates based on Lagrangian height-change measurements"

*Modified to "basal melt rates from Lagrangian Dh/Dt observations"*

399: "from A single, fixed"

*Changed to "fixed"*

400: "step but, in practice, ..." (move first comma).

*Removed "in practice"*

401: "useD", "substituteD"

*Fixed*

521: "downstream of THE grounding line"
*Fixed*

555: "over keels" is ambiguous; "on keels" or something else would be clearer.
*Changed to "on keels"*

577: Why quotation marks around "resolution" ?
*Removed quotation marks*

**Anonymous Referee #2**

This paper presents an extensive study of high-resolution satellite DEMs for determining elevation changes and ice-shelf basal melting of Pine Island Glacier in unprecedented detail. The amount of data and complexity of the analysis are impressive and well justified. A lot of care has been taken to find optimum methodologies and complementary ways of representing basal melt rates in a spatial domain that is constantly evolving. The high technical quality of the paper will make it a benchmark for related studies in the future, and the detailed results of basal melting will be highly useful for modellers.

The glaciological part of the paper is somewhat drowning in all the technicalities, but has a high potential for follow-up papers concerning links with ice-ocean interactions. It is therefore still well suited for a journal like the Cryosphere.

*We thank the reviewer for this favorable assessment. Indeed, a follow-on publication highlights interesting results on the temporal evolution of melt. We hope that the current paper will satisfy both technical readers and those who are more interested in the main results can skip the detailed methodology.*

The main things I would like to see improved before publication are the overly fragmented sections and the figures which are sometimes too excessive and not as informative as they could have been. More details on that are provided below, first on the figures and then on the manuscript text in line-by-line comments and edits which are mostly of minor nature.

*We reorganized and combined sections in the text and modified several figures, as described in the specific comments.*

**Figure comments and suggestions:**

Fig. 1: Although an impressive data set, I do not see the purpose of these regional DEM figures when the paper is all about PIG. Figure 2d would be a better opening figure, potentially with Fig. 1A or similar as an inset to show the location of PIG within West Antarctica. The present Fig. 1 could still be included in the Supplement, and in that case grounded ice and ocean areas could be masked out in panel b to highlight the ice shelves.

*We appreciate the reviewer's point, but Figure 1 provides context for Figure 2 and the PIG DEM time series. We also feel it is important to document the entire DEM dataset that was generated, to highlight potential for other ice shelves in the region.*

Fig. 2: The upper three panels could be split into a separate figure to better highlight the detailed and nice ice shelf one. The ice shelf outline is hardly visible in panels b and c, which should be improved for context, preferentially with bedrock and velocity close-ups of the ice shelves like the upper third panel of Fig. S1. That would be a good opportunity to highlight the

"transverse seabed ridge" in the main paper, as well as velocity variations on the main ice shelf which are totally saturated in Fig. 2c. In that case, a new multi-panel figure would definitely be needed and could replace the current Fig. 1.

*As suggested, we split Fig 2A-C and Fig 2D. The latter is now Figure 3 (with subsequent figure numbers updated). We added a label for "TSR" to show location of transverse seabed ridge on a figure in main text.*

Fig. 2 caption: I assume that it is over Bedmap2 surface hillshade, not "Bedmap2 DEM".

*This is the WorldView/GeoEye DEM composite embedded in the Bedmap2 DEM, with a shaded relief map derived from the combined product.*

For panel B; "...anisotropic interpolation of available data" (or be more specific).

*Changed to "Bed topography from anisotropic interpolation of radar-derived ice thickness (Section 2.6)"*

Unless grids or coordinates are included in any of the figures, the caption should say something about the location and map projection.

*Added "Projection is Antarctic polar stereographic (EPSG:3031)." to Figure 1 caption.*

Fig. 3: Could be moved to the supplement, but since this paper focuses so much on methodology it is okay to keep here. Please define in the caption what the dashed lines (mean or median of all errors?) and error bars represent (16-84% spread?). The small text at the top is somewhat cryptic and should be removed or included in the caption.

*We modified the caption as suggested. As stated in the caption, additional details on interpreting this figure can be found in (Shean et al. 2016).*

Fig. 4: This is a powerful illustration of the importance of careful co-registration for studies like this. To be improved: Make panel numbers as in preceding figures. Define what "Count" is. Include the PIG grounding line in white since that partly coincides with where the DEM variability is higher.

*We added panel letter labels and the grounding line, and modified the caption to better define count as per-pixel DEM count.*

Fig. 5: Panel numbers missing.

*We do not feel panel labels are needed for this figure, as years are listed. Defer to TC editorial staff.*

Fig. 6: This figure and the elaborate caption give a good explanation for a very complex methodology, but repeats and overlaps with the manuscript text which is unnecessarily fragmented into a series of sub-sections. These parts need to be better harmonized, potentially by moving much of the caption into the main text and removing repeated information from there.

*We acknowledge that this caption is long. But also acknowledge that many readers may not read detailed methodology in text. We prefer to keep the extended caption and detailed description in the text. As stated earlier, we reorganized these sections to alleviate fragmentation.*

Fig. 7: A shaded relief would be a nicer background than black in panels a and c. Ice shelf outlines (in white) would also be helpful in all panels to see relations with dh/dt patterns.

*We added the grounding line, as suggested by both reviewers. We experimented with a shaded relief background, but felt that it obscured many of the ICESat tracks with near-zero elevation change (white), so we kept solid black background to make sure that these tracks are visible.*

Fig. 8: Switch order of panels similar to the order of discussion in the text? I am not sure if the color scale is optimum, a lot of the ice shelf appear dark with no visual contrast.

*Good suggestion to switch panels for order of appearance. We experimented with several color stretches, and this one (with contours) was chosen to bring out as much detail as possible over the main shelf. The reviewer seems to be focused on small melt rates over the North and South shelves, which are small, within measurement uncertainty, and not the focus of this work.*

Fig. 9: It is a very good idea to separate channels and keels in the figure, but in the end it is actually difficult to see clear differences between panels and that also makes some of the main conclusions of the paper weaker. Have you tried other color bars or different scaling to see if differences may come out more clearly? Panel numbers are also missing.

*We disagree that it is difficult to see differences. And yes, we experimented with several different color stretches and settled on these to best capture the range of melt rates for channels/keels on both the inner and outer shelf.*

Fig. S1: Include panel numbers. The header "anisotropic interpolation" is confusing unless more details are provided (interpolation of what?).

*This is described in Section 2.6 of the main text, with a citation.*

Fig. S2: Something wrong with the reference here. I assume you want to refer to the similar Fig. 3 in the main manuscript. See also my comment to that figure.

*Yes, it should reference Figure 3 in the main text. It looks like this was a cross-reference that was not updated. Fixed.*

Fig. S4: See comment to Fig. 8. Although different colorbar stretches here are helpful C3 for interpretation of Fig. 8, I still feel that a lot of area appear dark without visible contrast.

*Addressed with earlier comment. It is impossible to choose a "perfect stretch" for this dataset. The stretches for the figures were carefully chosen to emphasize main points. We hope that any readers who are unsatisfied with the color ramp will download the data and visualize with their preferred stretch.*

**Line-by-line comments and edits:**

62-66: All the information and parentheses make this sentence hard to read. Split/rewrite or remove some content. This is also good to keep in mind elsewhere in the manuscript during revisions.

*We are following standard citation formatting for TC.  We feel that it is important to include these citations, and feel that this is the most concise way to do so.  If these were numbered superscript citations, as in a Science/Nature paper, this would be less of an issue.*

67: To be clear; "... across the grounding line of the South ice shelf"
*Fixed*

82: Fig. 2C is not a good reference here since it saturates at 1 km/yr. Remove.
*Removed*

83: The "South shelf" has already been introduced at L67, but more clearly here. Perhaps the order of the text should be switched or L67 can be simplified/removed.
*Yes, this is true, but we would rather provide historical context for PIG before getting into detailed geography for the study period. Changed to:*
*"...with a corresponding increase from ~10 to ~12 Gt/yr across the grounding line of the South PIG ice shelf (e.g., the "Wedge" catchment of Medley et al. (2014))"*

92: This is an important feature and therefore it is somewhat unfortunate that the reader is only referred to the supplement. However, it is also visible in Fig. 2B if the ice shelf outline is made thicker to highlight the relevant area, i.e. reference (Fig. 2B and S1).
*Changed to "Figure 2B and S1". We split figure 2 and added label for 2B.*

198: Four levels of sections is quite elaborate for a single research paper. I think that subdivision into 2.2.2.1 and 2.2.1.2 is more confusing than helpful.
*Reorganized to remove all level-4 headings*

228: I assume that this applies to the ice-shelf side of the grounding line. Suggestion for rewrite: "To provide a smooth transition from fully grounded ice to freely floating ice, we defined the coefficient ... downstream of the grounding line:"
*Good suggestion, changed as suggested.*

Eq. 3: t is not defined
*Modified to reiterate that t represents time*

257-259: It is not clear what the "tilt tolerance" is and how it was used.
*We changed "tilt tolerance" to "limits for tilt magnitude" - this is another set of weights in the least-squares optimization.*

267: Define or remove CE90/LE90
*These are 90th percentile of circular/linear error - common accuracy metrics in remote sensing literature.  Removed for clarity, as the reader can reference the Shean et al (2016) paper, where these terms are defined and discussed.*

335: Mention "anisotropic" here to make the link with Fig. S1 obvious.
*Good suggestion.  Added.*

345: How was the finer interpolation of the coarse RACMO data done?
*Changed to "We generated gridded RACMO SMB products with the same extent and spatial sampling as the DEM and velocity products using bicubic interpolation."*

352: How was firn air content removed? As a constant or time-variable?
*This is discussed in Section titled "Uncertainty and sources of error".  We added a sentence on this to address comment by Reviewer #1.*

353: Here you talk about surface accumulation, earlier SMB. Be consistent.
*Good catch.  Changed to SMB*

365: What do you mean by "dropping the constant d"? It is probably reasonable to treat it as a constant, but that should be briefly discussed/justified, e.g. in section 2.7.
*This was addressed in response to Reviewer #1.*

418: I think the word "product" is misleading here since it's more a description of how the calculation is done, whereas composites/mosaics are described later.
*Removed "products"*

443: Is a separate section needed? It is all almost about the initial-pixel approach, and the last sentence can be moved to line 417.
*We prefer a separate section that compares the two approaches and ties back to the end member scnerios described earlier.*
*We moved the last sentence to line 417, but it in the end, felt that it was redundant and removed entirely*

482: The simplified assumption of a constant firn air content in space and time is probably a minor issue for PIG, but not for most other ice shelves in Antarctica. A brief discussion or recapture of the cited reference would therefore be appropriate.
*Yes, and our +/- 2 m uncertainty should account for this spatial and temporal variability.  As this paper focuses on PIG, we do not feel it is necessary to comment on this issue for other ice shelves.  We look forward to future higher-resolution SMB and firn air content products that can further constrain this important correction for PIG and other shelves with lower basal melt rates.*

559: In this section you discuss characteristics of channel and keel melting shown in Fig. 9, but refer to arrows in Fig. 8. That is confusing, and even more so when some features like the "prominent longitudinal keel" is generally visible in Fig. 8, but not in Fig. 9 where keels and channels are separated.
*We believe that the arrows in Figure 8 are sufficient.  Figure 9 is a composite of all years, not a snapshot view of the shelf.  During the 2008 to 2015 period, the shelf experienced a rotation, as*

*documented by (Christianson et al., 2016; Jeong et al., 2016). As a result, the longitudinal channels/keels shifted their position on the Eulerian grid. When creating a weighted average composite of keels from all years, this feature is not as prominent due to this motion.*

593: It is very hard to actually see this in Fig. 9. Is it possible to find a better color scheme that makes this key point more convincing?
*We modified the color scale for Figure 9.*

656: FDM is not used in the manuscript as far as I can see (constant firn air content)
*Good catch. This was used in the* *(Shean et al. 2017)* *TC paper, but we removed any mention of FDM in this paper.*

Table 3: Define "n"
*Changed to "Minimum number of DEMs"*

References:

Christianson, K., Bushuk, M., Dutrieux, P., Parizek, B. R., Joughin, I. R., Alley, R. B., Shean, D. E., Abrahamsen, E. P., Anandakrishnan, S., Heywood, K. J., Kim, T.-W., Lee, S. H., Nicholls, K., Stanton, T., Truffer, M., Webber, B. G. M., Jenkins, A., Jacobs, S., Bindschadler, R. and Holland, D. M.: Sensitivity of Pine Island Glacier to observed ocean forcing: PIG response to ocean forcing, Geophys. Res. Lett., 43(20), 10,817–10,825, doi:10.1002/2016GL070500, 2016.

Jeong, S., Howat, I. M. and Bassis, J. N.: Accelerated ice shelf rifting and retreat at Pine Island Glacier, West Antarctica, Geophysical Research Letters, 43(22), 11,720–11,725, doi:10.1002/2016gl071360, 2016.

Joughin, I., Shean, D. E., Smith, B. E. and Dutrieux, P.: Grounding line variability and subglacial lake drainage on Pine Island Glacier, Antarctica, Geophysical Research Letters, 43(17), 9093–9102, doi:10.1002/2016gl070259, 2016.

Shean, D. E., Alexandrov, O., Moratto, Z. M., Smith, B. E., Joughin, I. R., Porter, C. and Morin, P.: An automated, open-source pipeline for mass production of digital elevation models (DEMs) from very-high-resolution commercial stereo satellite imagery, ISPRS Journal of Photogrammetry and Remote Sensing, 116, 101–117, doi:10.1016/j.isprsjprs.2016.03.012, 2016.

Shean, D. E., Christianson, K., Larson, K. M., Ligtenberg, S. R. M., Joughin, I. R., Smith, B. E., Max Stevens, C., Bushuk, M. and Holland, D. M.: GPS-derived estimates of surface mass balance and ocean-induced basal melt for Pine Island Glacier ice shelf, Antarctica, The Cryosphere, 11(6), 2655–2674, doi:10.5194/tc-11-2655-2017, 2017.